# Overflow Prevention Enhances Long-Context Recurrent LLMs

**Assaf Ben-Kish[1], Itamar Zimerman[1,2], M. Jehanzeb Mirza[3],**
**Lior Wolf[1], James Glass[3], Leonid Karlinsky[4], Raja Giryes[1]**
[1]Tel Aviv University, [2]IBM Research, [3]MIT CSAIL, [4]Xero

https://github.com/assafbk/OPRM

## Abstract

A recent trend in LLMs is developing recurrent sub-quadratic models that improve long-context processing efficiency. We investigate leading large long-context models, focusing on how their fixed-size recurrent memory affects their performance. Our experiments reveal that, even when these models are trained for extended lengths, their use of long contexts remains underutilized. Specifically, we demonstrate that a chunk-based inference procedure, which identifies and processes only the most relevant portion of the input can mitigate recurrent memory failures and be effective for many long-context tasks: On LongBench, our method improves the overall performance of Falcon3-Mamba-Inst-7B by 14%, Falcon-Mamba-Inst-7B by 28%, RecurrentGemma-IT-9B by 50%, and RWKV6-Finch-7B by 51%. Surprisingly, this simple approach also leads to state-of-the-art results in the challenging LongBench v2 benchmark, showing competitive performance with equivalent size Transformers. Furthermore, our findings raise questions about whether recurrent models genuinely exploit long-range dependencies, as our single-chunk strategy delivers stronger performance - even in tasks that presumably require cross-context relations.

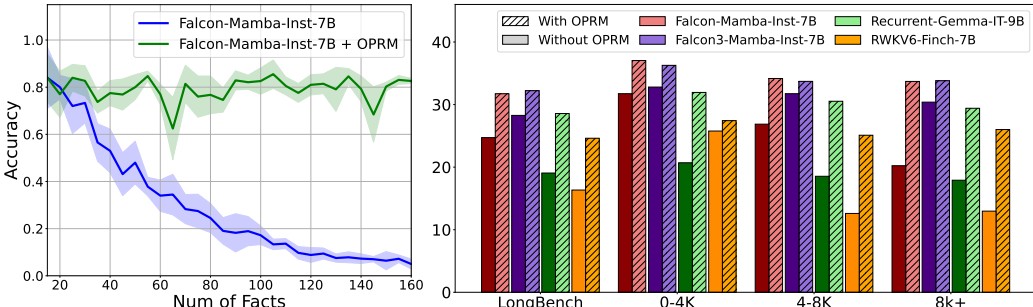

Figure 1: **Limited recurrent memory capacity limits leading long-context LLMs**.
(**Left**) We quantify this behavior by measuring zero-shot associative recall curves: The x-axis represents the number of facts (key-value pairs) in the context, while the y-axis shows the retrieval accuracy of the correct value from the context. To mitigate the memory problem, we propose OPRM, a chunk-based inference strategy that does not force the model to encode more information than it can reliably store. (**Right**) Leading recurrent LLMs evaluated on LongBench and LongBench_e ('0-4K', '4-8K', '8K+'). Surprisingly, our simple approach significantly improves performance on long-context tasks.

## 1 Introduction

Long sequences, which can span millions of tokens - such as entire books, lengthy dialogues, code repositories, or genomic data - commonly arise in real-world applications. While Transformers (Vaswani et al., 2023) have achieved remarkable results on short-sequence tasks, their applicability to long-sequence scenarios is hindered by the quadratic cost of self-attention. In response, researchers have explored sub-quadratic recurrent large language models (LLMs) that process extended inputs more efficiently (Gu & Dao, 2024). Despite their promise, existing recurrent LLMs like Falcon3-Mamba-Inst-7B (Team, 2024) often fall

short of their full potential on long-context tasks. Notably, length generalization alone does not explain this limitation (Ben-Kish et al., 2025), nor it is fully resolved by context-extension approaches (Azizi et al., 2025; Ye et al., 2025), given that these models are already trained on sequence lengths that surpass typical long-context benchmarks.

We hypothesize that the key bottleneck lies in how relevant information is captured and utilized across the entire input. While previous studies utilize toy problems such as needle in a haystack to study the limitations of recurrent models (Ben-Kish et al., 2025), here we utilize the Associative Recall (AR) task to explore their limitations. We find that even very large hidden states, such as the one of Falcon-Mamba-Inst-7B (Team, 2024), have a relatively limited memorization capacity. We quantify this phenomenon with the AR measurements, which uncover an overflow-like behavior that occurs even for short sequence inputs. This simple setting suggests that even if recurrent LLMs accept extended sequences, they do not fully leverage all the information in the context.

Based on this observation, we propose a chunk-based inference strategy that preemptively segments the input context, letting the model process each segment in parallel and then select the most relevant chunk for decoding. This simple procedure effectively avoids memory overflows, as it never forces the model to encode more information than it can reliably store at once. Surprisingly, it boosts recurrent LLMs' performance on a variety of long-context tasks, e.g., in LongBench (Bai et al., 2024). In particular, it achieves state-of-the-art (SOTA) results on the challenging LongBench v2 (Bai et al., 2025) benchmark, beating Transformer counterparts of similar scale, while maintaining sub-quadratic complexity.

Beyond improving standard long-context tasks, this chunk-based framework naturally extends the model's usable context length without requiring additional training. Whereas existing context-extension methods often introduce complexity or compromise performance, our approach integrates directly into the inference pipeline and handles sequences far longer than those seen at training time. By adjusting chunk sizes according to the information density at hand, we mitigate memory constraints in practice, thereby enabling recurrent LLMs to operate effectively even at very large scales.

Our contributions are: (i) We uncover how large recurrent LLMs often suffer from chronic underuse of available in-context information - despite being trained on long contexts and having very large hidden states. (ii) We present a chunk-based inference method that eases memory overflows and significantly improves results across diverse benchmarks. (iii) We show that this simple, training-free technique yields improvements in context extension, enabling recurrent LLMs to match or exceed leading Transformer-based models on long context tasks. (iv) Crucially, these results raise questions about whether recurrent LLMs truly capture long-range dependencies across widely separated parts of the input, as our single-chunk approach improves their performance on a large variety of long context tasks without using cross-segment information.

## 2 Related Work

LLMs are primarily built on the Transformer architecture. However, their quadratic complexity in sequence length poses a significant challenge for long-context applications. To address this, a promising recent trend is the development of recurrent LLMs, which offer improved efficiency. During the prefill phase, these models exhibit sub-quadratic complexity in sequence length, making them more efficient than the quadratic complexity of Transformers. In auto-regressive decoding, their recurrent structure allows for $O(1)$ complexity per step, outperforming the linear complexity of optimized Transformers that implement KV caching.

Scaling these architectures has shown competitive performance with SOTA Transformers on standard short-context benchmarks, particularly at model sizes exceeding 7B parameters, marking a breakthrough in language modeling. Notable examples include Falcon-Mamba (Zuo et al., 2024), Eagle 7B (Peng et al., 2024), xLSTM-7B (Beck et al., 2025), Hawk (De et al., 2024), and others (Waleffe et al., 2024). We turn to elaborate on these recurrent models.

**Mamba** layers are built on an evolved form of state-space layers introduced by Gu et al. (2021b;a). They exhibit promising results across several domains, including NLP (Waleffe et al., 2024; Wang et al.), audio (Miyazaki et al., 2024), image processing (Zhu et al., 2024; Liu

et al., 2024b), RL (Lv et al., 2024), and more (Behrouz & Hashemi, 2024). Recent works scaled Mamba up to 7B parameters (Zuo et al., 2024; Team, 2024). The most advanced models, Falcon-Mamba and Falcon3-Mamba, were trained on 5.5T/7T tokens with sequence lengths of up to 8K/32k tokens, respectively. These models exhibit Transformer-level performance on short-sequence tasks, while enjoying the efficiency of the Mamba layer. A full definition of Mamba can be found in Appendix E.3.

**RWKV** (Peng et al., 2023) is a linear attention variant that replaces dot-product attention with channel-directed attention, enabling a recurrent form. It matches SoTA comparable-size Transformers in both NLP and vision (Duan et al., 2025; Fei et al., 2024).

**Hybrid models** combine recurrent and attention layers to enhance both the effectiveness and efficiency of LLMs (Poli et al., 2024; Team et al., 2024b; Dong et al., 2024) demonstrating remarkable performance across multiple domains, including NLP (Ren et al., 2024; Lenz et al., 2025), vision (Hatamizadeh & Kautz, 2024), RL (Huang et al., 2024), etc. Yet, as they include attention layers, they still suffer from quadratic complexity w.r.t the input context length. Some hybrid models avoid the quadratic cost by replacing dense attention with local attention (Ren et al., 2024; Arora et al., 2024), hence enjoying sub-quadratic complexity.

**RecurrentGemma.** The Griffin architecture (De et al., 2024) demonstrated that a combination of gated linear recurrent units (Orvieto et al., 2023) and local attention layers can outperform SoTA Transformers at the 7B scale, while being more efficient. Scaling Griffin led to the RecurrentGemma models (Botev et al., 2024) that matched the performance of Transformer-based Gemma (Team et al., 2024a) while being trained on less tokens.

**Fixed Memory Capacity in Recurrent LLMs.** Arora et al. (2024) shows that the recurrent memory capacity of the model increases as its state size grows. Guided by this principle, recent works increase the hidden state's size in order to improve performance (Dao & Gu, 2024; Beck et al., 2024; Qin et al., 2024b; Arora et al., 2024). Yang et al. (2025); Sun et al. (2025) propose recurrent layers with improved update mechanisms for better hidden state utilization. While these approaches increase memory capacity or manage it more effectively, it still remains bounded, hence may suffer from overflows. Moreover, because the capacity is fixed, re-training with a larger state may be required for certain downstream tasks.

**Long-Context Capabilities.** Despite their promising efficiency, SOTA recurrent and hybrid LLMs with sub-quadratic complexity have yet to match the performance of leading Transformer-based models in the regime they are designed for: real-world long-range tasks. We show that this limitation arises from their fixed memory capacity, which makes them prone to overflows, and provide a simple approach to mitigate this behavior.

**Additional related work** on RAG-based methods, as well as context-extension methods for recurrent LLMs such as DeciMamba (Ben-Kish et al., 2025), is provided in Appendix E.

## 3 Problem Investigation

We use the AR task (described below) to investigate the memory overflow phenomenon in SoTA recurrent LLMs. Memory overflows are a failure mode where the model cannot store or retrieve relevant information from the context. Specifically, when relevant information is combined with only a small amount of additional information (e.g., a few extra facts), the model can retain and retrieve it successfully. However, as more additional information is introduced, the effective state capacity is exceeded, resulting in retrieval failures. In the AR task, overflows are indicated by a drop in accuracy relative to the initial value.

**Associative Recall (AR).** In psychology, Associative Recall (AR) is defined as the ability to learn and remember the relationship between unrelated items. In the context of LLMs, AR plays an important role in the emergence of critical learning capabilities (Olsson et al., 2022), hence widely studied (Arora et al., 2025; Lutati et al., 2023; Okpekpe & Orvieto, 2025). The AR task that we use in our experiments is similar to Arora et al. (2023): The context is a concatenation of $M$ facts, which are key-value pairs $(K_i, V_i)$, $i \in [M]$, and are eventually followed by a query $Q$, which is equal to one of the keys in the context:

AR Sample: $K_1\ V_1\ <pad>\ <pad>\ K_2\ V_2\ <pad>\ <pad>\ \cdots\ K_M\ V_M\ Q$ **?**

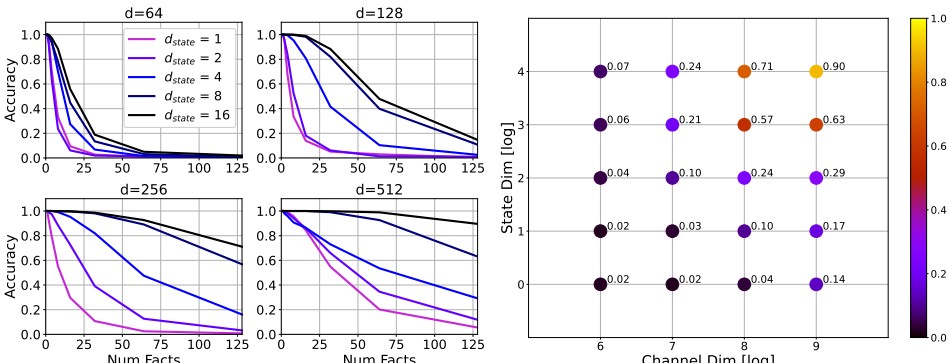

Figure 2: **Memory overflows in a controlled setup**. (**Left**) AR curves of 2-layer models with different hidden state sizes. $d$ is the channels dimension and $d_{state}$ is the state size. The x-axis represents the number of facts in the context, and the y-axis shows the retrieval accuracy of correct values from the context. (**Right**) Memory capacity as a function of channel and state dimensions. Each point shows the ratio between the capacity (maximal amount of retrieved facts from a single context) and the amount of facts that the model was trained to retrieve (here, 128 facts). All 2-layer models were trained to retrieve up to 128 facts, yet most models suffer from overflows, therefore have a lower capacity than the one trained for.

The keys and values are drawn from a token vocabulary $\mathbb{V}$. In our experiments each key or value is between 1 to 3 tokens long, depending on the experiment. The objective of the model is to predict the corresponding value for the query $Q$. It is important to note that there are no contradictions between the facts, i.e. a key $K_i$ cannot be attributed to two different values. Lastly, to control for the effects of varying context lengths, all contexts are padded to a pre-defined length $N$, using special pad tokens that are inserted between key-value pairs. Thus, all samples have the same length, regardless of the amount of facts in the context.

**Zero-Shot AR with SoTA Recurrent LLMs.** We evaluate a recurrent model with $M \in [15, 160]$ facts, where each context is padded to a length of $N = 1200$ tokens. Implementation details are provided in Appendix A.1. Our results are displayed in Figure 1 (left, blue curve). The graph shows the behavior of a popular recurrent LLM, Falcon-Mamba-Inst-7B, which has a relatively large hidden state size: 4096 channels, each with a state size of 16. The AR profile behaves as follows: When the number of facts is small, we observe a healthy memory behavior, achieving slightly higher than 80% accuracy. Then, as we increase the amount of facts in the context, the performance starts to drop, revealing the first occurrences of overflows. When the amount of information is increased further, the model remembers only a few single facts out of the hundreds in the context. This measurement is performed with short contexts ($N = 1200$), emphasizing the severity of this issue in the long-context domain. In Section B.5 (Appendix), we measure the sensitivity of AR performance to the input sequence length $N$. We find that the model is not particularly sensitive to the context's length, but rather to the amount of information it contains. This further shows that overflows are a fundamentally distinct limitation of recurrent LLMs - one that should be addressed independently of other known limitations, such as length generalization. Lastly, Section B.6 (Appendix) presents an analysis of how overflows vary with token position.

**AR in a Controlled Setup.** To strengthen the validity of this result, we capture the overflow phenomenon in an additional independent, controlled, setup where we train 2-layer Mamba models from scratch on a synthetic AR task. We test different combinations of channels dimension $d \in \{64, 128, 256, 512\}$ and state dimension $d_{state} \in \{1, 2, 4, 8, 16\}$, resulting in a wide variety of hidden state sizes (In Mamba, defined by $d \times d_{state}$). In addition, we use contexts of length N=384 tokens, and train each model to retrieve up to 128 facts. More implementation details are provided in Appendix A.2. As can be seen in Figure 2 (left), the 2-layer models produce an associative recall profile similar to that produced by the SoTA recurrent LLM in the zero-shot setting in Figure 1 (left, blue curve). Furthermore, when increasing the hidden state size, the overflow phenomenon becomes less severe, yet, it is not fully prevented. This highlights the problem with the 'fixed memory scaling' approach discussed in Section 2: vanilla recurrent models will never be able to process contexts of

arbitrary data content, such as large code repositories or long videos. We visualize this in Figure 2 (right), where each data point shows the memory capacity for a given hidden state size. While the capacity increases with state size, the models are still not able to retrieve all 128 facts from the context, despite being trained to do so. As for the scaling trends, we find that both channel and state dimensions are necessary for increasing the capacity - yet the amount of channels must be significantly larger than the amount of states - here $\times 32$.

## 4 Method

From Section 3, we conclude that for given channel and state dimensions, the maximum capacity of the recurrent memory is limited. Furthermore, the performance of the model degrades as the amount of information increases beyond this limit. Motivated by this observation, we propose OPRM (Overflow Prevention for Recurrent Models), a simple yet effective inference method. The core idea is to chunk the context such that the information content of each chunk does not exceed the model's limit. Surprisingly, we find that decoding a single relevant chunk yields significant gains over the baseline across a variety of tasks.

**Overall Mechanism.** We consider a generation task where a prompt $X = [P, C, S]$, consists of a prefix $P$, a context $C$, and a suffix $S$, which contains a query $Q$. The model is then given $X$ to generate an answer $A$. Our method operates in two stages: **speculative prefill** and **selective decoding**, which are described in the next two subsections (Secs. 4.1 and 4.2). This design follows the standard prefill-decode framework in LLMs, as visualized in Figure 3, and formally described in Algorithm 1. Our method is grounded in several key design principles, including the role of locality and compression in NLP, as well as a speculative processing strategy. Further details on these aspects are provided in Appendix C.

---

**Algorithm 1** Overflow Prevention for Recurrent Models (OPRM)

**Input:** Prompt $X = [P, C, S]$, recurrent model $\mathcal{M}(H, X)$ with state $H$ and input $X$, chunk size $L$, decoding algorithm $\tau$ (e.g. - greedy)
**1. Preprocessing - prepare OPRM chunks:**
  1.1 Right pad $C$ s.t. $|C| \bmod L = 0$; $b \leftarrow |C|/L$
  1.2 $\forall i \in [b]$: $C_i \leftarrow C[(i-1) \cdot L : i \cdot L]$
  1.3 $\forall i \in [b]$: $X_i \leftarrow [P, C_i, S]$
**2. Speculative Pre-Fill:**
  2.1 $\forall i \in [b]$ (in parallel): $H_i, Pr(\cdot \mid X_i) \leftarrow \mathcal{M}(0, X_i)$   ▷ Compute state and logits for all chunks
**3. Selective Decoding:**
  3.1 $\Gamma_{IDK} \leftarrow \{\arg\max\{Pr(v \mid X_i) \mid v \in \mathcal{V}\} = \text{error\_token\_id} \mid \forall i \in [b]\}$   ▷ Find IDK Chunks
  3.2 $j \leftarrow \arg\min\{E_i \mid i \in [b]/\Gamma_{IDK}\}$   ▷ Select decoding state (here - entropy criteria)
  3.3 $A_{j,0} \leftarrow \tau(Pr(\cdot \mid X_j))$   ▷ Obtain first answer token
  3.4 $A \leftarrow \tau(\mathcal{M}(H_j, A_{j,0}))$   ▷ Decode answer $A$ autoregressively starting from state $H_j$
**return** $A$

---

### 4.1 Speculative Prefill

OPRM first splits the context C into b chunks $\{C_i\}_{i=1}^{b}$ of the same length $L$ and constructs b separate prompts, each maintaining the original structure: $X_i = [P, C_i, S]$. These prompts are processed in parallel in a speculative manner, efficiently computing the output distribution for the first answer token $Pr(\cdot \mid X_i)$, the state after processing $X_i$ (denoted as $H_i$), and the token predicted based on $Pr(\cdot \mid X_i)$ and decoding algorithm $\tau$ (denoted as $A_{i,0}$).

### 4.2 Selective Decoding

During decoding, we select the state $H_j$ and token $A_{j,0}$ from the most informative prompt $X_j$ based on a selection criterion, and then perform auto-regressive decoding. We propose two criteria for selecting the most relevant chunk, $X_j$:

(i) **Entropy-based Criteria:** Following Malinin & Gales and Yona et al., we quantify uncertainty by computing the entropy of the output distribution for each chunk $X_i$:

$$j = \arg\min_i\{E_i \mid i \in [b]\}, \quad E_i = \sum_{v \in \mathcal{V}} Pr(v \mid X_i) \cdot \log_2 Pr(v \mid X_i),$$

where $\mathcal{V}$ represents the vocabulary, and $E_i$ serves as an uncertainty score for each prompt.

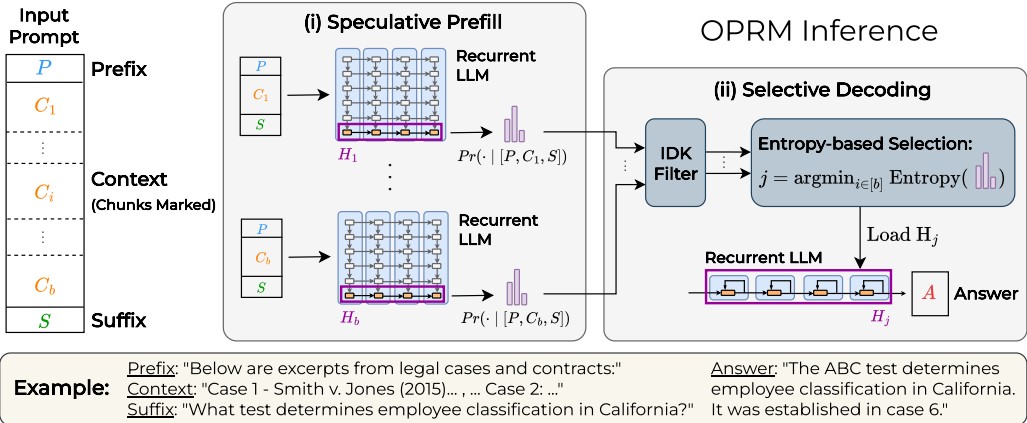

Figure 3: **Visualization of OPRM:** Given a prompt structured as Prefix ($P$), Context ($C$), and Suffix ($S$), we first split the context into chunks ($C_1, \cdots, C_b$). All chunks are wrapped with the Prefix ($P$) and Suffix ($S$) and are processed independently in a speculative manner (in parallel). During decoding, tokens are predicted auto-regressively, conditioned exclusively on the selected chunk.

(ii) **Probability-based Criteria**: Following Jiang et al. (2021), instead of relying on entropy, this criterion selects the chunk that maximizes the probability of the query tokens in the suffix. Given the output distribution $Pr(Q \mid [P, C_i])$, we choose the chunk $C_j$ that maximizes the likelihood of the query Q:

$$j = \arg\max_i \{P_i \mid i \in [b]\}, \quad P_i = Pr(Q \mid [P, C_i])$$

By prioritizing the chunk that yields the most confident prediction, we aim to select prompts with greater relevance between the query and the context.

**IDK Filter.** While the selection criterion returns the chunk with the highest confidence, it does not account for cases where the model is confident that it does not know the answer. This is important because in many tasks some chunks will not hold relevant information, and we do not want the model to select them. To address this, we employ a simple filtering technique using an IDK (I Don't Know) token. First, we add the sentence 'If the answer does not exist in the passages, return "Error"' to the suffix S. After the speculative pre-fill phase, we discard all chunks that predict the "Error" token. If all chunks predict the "Error" token, we retain only the first one. Finally, we apply the selection criterion to the remaining chunks and decode the selected one.

**Chunking Strategy.** While more complex chunking strategies exist, we find that using a fixed chunk size L, treated as a hyperparameter, is sufficient across tasks. Keeping the chunk size constant eliminates the need for sophisticated algorithms for state capacity estimation and is both easy to use and parallelize. By reducing the context size from $|C| = Lb$ to $|C_i| = L$, we effectively obtain a simple yet efficient method to mitigate memory overflows.

Appendix D provides details on additional advantages of OPRM, including an analysis of its improved complexity, insights into efficient computation, the role of the chunk size (the hyperparameter $L$) in controlling the memory-recall tradeoff, and other relevant aspects.

## 5 Experiments

We evaluate OPRM as an enhancer of long-context abilities of recurrent LLMs across multiple tasks, detailed in Sections 5.1 and 5.2. We then conduct ablation studies in Section 5.3, followed by an empirical analysis of our method's efficiency in Section 5.4. Additional experiments are provided in Section B (Appendix): comparisons to agentic and RAG-based long-context methods, more long context benchmarks (Zhang et al., 2024), and ablations exploring multi-chunk aggregation. In all experiments we perform OPRM inference with the min entropy selection criterion. All implementation details are reported in Appendix A.

| Model Type | Model | LB_v2 | Difficulty | | Length | |
|---|---|---|---|---|---|---|
| | | | Easy | Hard | 0-32k | 32k-128k |
| Recurrent | RWKV6-Finch-7B | 16.3 | 16.5 | 16.2 | 13.5 | 20.3 |
| | RWKV6-Finch-7B + OPRM | **22.9** | **22.0** | **23.4** | **17.4** | **30.9** |
| | Falcon-Mamba-Inst-7B | 2.8 | 3.5 | 2.4 | 3.4 | 2.1 |
| | Falcon-Mamba-Inst-7B + OPRM | **27.7** | **27.0** | **28.2** | **27.5** | **28.0** |
| | Falcon3-Mamba-Inst-7B | 24.0 | 20.0 | 26.2 | 22.5 | 25.9 |
| | Falcon3-Mamba-Inst-7B + OPRM | **30.2** | **33.0** | **28.6** | **28.7** | **32.2** |

Table 1: **LongBench v2 - comparison to baseline (short subset, samples < 150K tokens).** Results for leading recurrent LLMs with and without OPRM. Due to hardware constraints, results are shown for samples with less than 150k tokens. LB_v2 is the LongBench v2 score. Length is the length group in words. The length group '128k+' is not a part of the subset.

### 5.1 Long-Context evaluations

**Zero-Shot Associative Recall.** We repeat the experiment in Section 3 with OPRM inference. Our results for Falcon-Mamba-Instruct-7B can be seen in Figure 1 (left). While the baseline (blue) suffers from overflows quite early, with OPRM inference (green), the accuracy is invariant to the amount of information in the context, practically solving the task.

**LongBench.** LongBench (Bai et al., 2024) is an extensive real-world long-context benchmark that combines 16 different tasks across 6 categories: Single-Document QA, Multi-Document QA, Summarization, Few-Shot Learning, Synthetic Tasks, and Code Completion. Here, most models do not suffer from limited length generalization, as they were trained on contexts of same or longer length. Results for leading recurrent LLMs with and without OPRM inference are presented in Figure 1 (right). Full results can be found in Tables 13, 14, 15, 16 in the Appendix. By allocating a different state per chunk, OPRM is able to mitigate memory overflows. This change proves highly effective, as OPRM improves performance in almost all categories. The total improvement when applying OPRM inference is 14% for Falcon3-Mamba-Inst-7B, 28% for Falcon-Mamba-Inst-7B, 50% for Recurrent-Gemma-IT-9B, and 51% for RWKV6-Finch-7B. Additionally, we observe that as context length increases, the advantage of OPRM becomes more evident (4-8k, 8k+). This is very reasonable, as the amount of information in the context correlates with its length. Lastly, as shown in Table 12 (Appendix), we find that OPRM significantly improves multi-hop reasoning performance, which is more than doubled on certain benchmarks across all evaluated models. These results suggest that memory overflows severely degrade multi-hop reasoning capabilities, and that OPRM provides an efficient and effective solution.

**LongBench v2.** LongBench v2 (Bai et al., 2025) evaluates LLMs on a variety of real-world long-context tasks, with contexts ranging from 8K to 2M words. The tasks include single and multi-document QA, long in-context learning, long-dialogue, code understanding, and long-structured data. All questions are in the form of multiple-choice with four answers. A comparison of leading recurrent LLMs with and without OPRM inference is presented in Table 1. Here, we report performance for samples with less than 150K tokens (60% of the LongBench v2 dataset), as the baseline models suffer from Out-Of-Memory (OOM) errors for longer samples. Moreover, due to the long context, some of the responses provided by Falcon-Mamba-Inst-7B and RWKV6-Finch-7B are not valid. OPRM improves inference results significantly. Falcon3-Mamba-Inst-7B + OPRM achieves a score of 30.2, an improvement of 25.8%. Falcon-Mamba-Inst-7B + OPRM achieves 27.7, a significant improvement given that the baseline could not even predict one of the answers. RWKV performance also improved, as most of its predictions are now valid. Recurrent-Gemma-IT-9B has OOM in most samples; hence we do not report its results.

In Table 2 we compare leading 7 billion parameter Attention-based LLMs, all with a claimed context length of 128k tokens, and recurrent LLMs augmented with OPRM inference. For completeness, we also present larger Transformers (in gray). Here, we evaluate over the whole dataset (possible thanks to OPRM's chunking approach). We find that with OPRM inference, Falcon3-Mamba-Inst-7B and Falcon-Mamba-Inst-7B are competitive with equivalent-sized Transformer-based models. Falcon3-Mamba-Inst-7B + OPRM is even able to achieve a score of 30.8, setting a new SOTA result for this size class. Furthermore, while

Transformer-based LLMs perform better on tasks with contexts of 0-32K words, recurrent architectures with OPRM inference begin to gain an advantage as context lengths increase (32K-128K, 128K+). This trend persists even when context extension methods are applied to Transformer models (e.g., Qwen-2.5-Inst-7B + YaRN). Most importantly, the recurrent architectures are able to achieve this performance while having highly favorable efficiency.

| Model Type | Model | #Params | LB_v2 | Difficulty | | Length | | |
|---|---|---|---|---|---|---|---|---|
| | | | | Easy | Hard | 0-32k | 32k-128k | 128k+ |
| – | Random Chance | – | 25.0 | 25.0 | 25.0 | 25.0 | 25.0 | 25.0 |
| | Human | – | 53.7 | 100.0 | 25.1 | 47.2 | 59.1 | 53.7 |
| Transformer (Large) | Llama-3.1-Inst-70B | 70B | 31.6 | 32.3 | 31.2 | 41.1 | 27.4 | 24.1 |
| | Mistral-Large-Inst-2411 | 123B | 34.4 | 38.0 | 32.2 | 41.7 | 30.7 | 29.6 |
| | Qwen-2.5-Inst-72B | 72B | 39.4 | 43.8 | 36.7 | 44.4 | 34.0 | 41.7 |
| | Qwen-2.5-Inst-72B + YaRN | 72B | 42.1 | 42.7 | 41.8 | 45.6 | 38.1 | 44.4 |
| Transformer (Medium) | Llama-3.1-Inst-8B | 8B | 30.0 | 30.7 | 29.6 | 35.0 | 27.9 | 25.9 |
| | GLM-4-Chat-9B | 9B | 30.2 | 30.7 | **29.9** | 33.9 | 29.8 | 25.0 |
| | Qwen-2.5-Inst-7B | 7B | 27.0 | 29.2 | 25.7 | 36.1 | 23.7 | 18.5 |
| | Qwen-2.5-Inst-7B + YaRN | 7B | 30.0 | 30.7 | 29.6 | **40.6** | 24.2 | 24.1 |
| Hybrid | RecurrentGemma-IT-9B + OPRM | 9B | 26.2 | 26.0 | 26.4 | 26.1 | 22.8 | **33.3** |
| Recurrent | RWKV6-Finch-7B + OPRM | 7B | 22.7 | 16.5 | 16.2 | 18.3 | 27.0 | 21.3 |
| | Falcon-Mamba-Inst-7B + OPRM | 7B | 29.4 | 30.2 | 28.9 | 27.8 | 31.2 | 28.7 |
| | Falcon3-Mamba-Inst-7B + OPRM | 7B | **30.8** | **34.4** | 28.6 | 29.4 | **32.6** | 29.6 |

Table 2: **LongBench v2 - Comparison to SOTA models (all samples).** Results for leading recurrent LLMs with OPRM Inference, along with leading Attention-based LLMs. LB_v2 is the LongBench v2 score. Large open-source models are added in gray. In the top two rows we show the Random Chance and Human scores, as provided by the LongBench v2 paper.

## 5.2 Context Extension

Here, we focus on models that were trained on short sequences, and evaluate them on long sequences. As OPRM naturally performs context extension, we compare it to dedicated methods, and show that the latter are less effective, as they do not prevent overflows. An additional Needle-In-A-Haystack extension experiment is provided in Appendix B.3.

**LongBench.** Following Ye et al. (2025), we use a Mamba-1.4b model that was trained with 2k token sequences, which are shorter than the ones in the benchmark. We compare various context extension methods, including OPRM, on all 6 LongBench_e categories. The results are in Table 3. We find that OPRM beats the dedicated extension methods, sometimes even by a large margin. This can be explained by their objective: While they attempt to increase the amount of tokens that the model can process, they do not account for memory capacity, hence do not prevent overflows. In contrast, OPRM benefits from length generalization, as preventing overflows naturally extends the context. We note that the MambaExtend authors (Azizi et al., 2025) did not share their code, hence we could not compare with them.

| Method | SD-QA | MD-QA | Summ | Few-Shot | Syn | Code | Avg |
|---|---|---|---|---|---|---|---|
| Mamba-1.4b | 5.94 | 5.92 | 7.80 | 12.56 | 3.00 | 12.92 | 9.35 |
| DeciMamba-1.4b | 6.42 | 6.19 | 9.78 | 17.54 | **3.12** | 35.17 | 15.25 |
| LongMamba-1.4b | 6.76 | 7.57 | 10.34 | 28.21 | 2.88 | 42.78 | 17.33 |
| Mamba-1.4b + OPRM | **11.36** | **8.92** | **20.22** | **35.58** | 2.60 | **43.25** | **21.22** |

Table 3: **Context Extension - LongBench_e.** We compare OPRM to leading context extension methods. We use a Mamba-1.4b model which was trained on 2K token sequences, which is shorter than most of the samples. SD-QA, MD-QA, Summ, Syn, Code, Avg stand for Single Document QA, Multi-Document QA, Summarization, Code Completion and Average. OPRM outperforms existing methods, highlighting its benefit during extension.

**Document Retrieval.** In this task, the model receives a query and $N_{docs}$ documents, with the objective of returning the ID of the document that contains the answer to the query. Our data is sampled from SQuAD v2 (Rajpurkar et al., 2018). We train Mamba and DeciMamba models on sequences of $N_{docs} = 11$ documents (around 2K tokens) and evaluate them with $N_{docs} \in [11, 240]$ (2K to 50K tokens). The results are presented in Table 4. Here again,

OPRM outperforms DeciMamba, the dedicated context extension method. This is noticeable especially in the larger model - showing that the overflow-prevention approach scales better.

| Model / # Docs | 10 | 20 | 30 | 40 | 50 | 60 | 80 | 100 | 120 | 140 | 160 | 180 | 200 | 240 |
|---|---|---|---|---|---|---|---|---|---|---|---|---|---|---|
| Mamba-1.4b | **88.0** | **88.0** | 76.7 | 52.0 | 31.0 | 23.3 | 7.3 | 5.7 | 3.3 | 0.3 | 0.3 | 0.0 | 0.7 | 0.0 |
| DeciMamba-1.4b | **88.0** | **88.0** | 79.3 | 60.0 | 41.3 | 24.7 | 11.7 | 2.7 | 1.7 | 1.3 | 0.3 | 0.0 | 0.3 | 0.0 |
| Mamba-1.4b + OPRM | **88.0** | **88.0** | **81.0** | **77.3** | **79.7** | **77.0** | **67.0** | **63.7** | **72.7** | **61.7** | **59.3** | **58.3** | **54.0** | **56.3** |
| Mamba-130m | **68.3** | 74.0 | 70.3 | 64.7 | 59.3 | 45.3 | 24.7 | 5.7 | 1.0 | 0.3 | 0.3 | 0.0 | 0.0 | 0.0 |
| DeciMamba-130m | 67.7 | **77.3** | **72.0** | **68.7** | **64.7** | **65.3** | 49.7 | 37.0 | 26.3 | 16.7 | 5.3 | 3.0 | 4.3 | 2.0 |
| Mamba-130m + OPRM | **68.3** | 74.0 | 68.7 | 62.0 | 57.0 | 60.3 | **56.3** | **56.3** | **53.3** | **53.7** | **45.3** | **48.3** | **44.0** | **43.0** |

Table 4: **Context Extension - Multi-Document Retrieval**. We show the scores of Mamba, DeciMamba, and Mamba + OPRM models as we increase the amount of documents during evaluation. We report the retrieval accuracy for 130M and 1.4B models. Both Mamba and DeciMamba models were trained using 11 documents. We find that OPRM is able to extend the context significantly, and scales well with model size.

## 5.3 Ablations

We ablate the components of OPRM below.

**Chunk Selection Method.** We test different methods on the HotPotQA benchmark (LongBench_e) with Falcon-Mamba-Inst-7B using $L = 3000$. Besides min entropy and max $Pr(Q \mid [P, C_i])$ selection we also show random selection and vanilla inference, all with a Falcon-Mamba-Inst-7B model. The results are in Table 5. Although maximizing $Pr(Q \mid [P, C_i])$ seems reasonable, it barely outperforms the random method. We find that this approach is highly unstable - empirically, the probability of the query almost always equals zero. This

| Method | 0-4K | 4K-8K | 8K+ |
|---|---|---|---|
| Baseline | 36.35 | 21.18 | 18.4 |
| Random | 35.53 | 23.02 | 27.62 |
| Max $Pr(Q \mid [P, C_i])$ | 37.14 | 26.13 | 25.76 |
| Min Entropy (Ours) | **39.71** | **37.1** | **35.18** |

Table 5: **Chunk selection method ablation.** We test Falcon-Mamba-Inst-7B using $L = 3000$ over HotPotQA (LongBench_e). Min Entropy better extends to longer contexts w.r.t other methods. Surprisingly, random select is better than the baseline, showing the severity of memory overflows.

can be explained by the probability computation: with a query typically 20-50 tokens long, the product of small numbers becomes very small, and any zero in the sequence makes the entire product zero. Next, we see that the baseline is outperformed by the random method as input length increases. Since the optimal chunk size for this task is $L = 3000$ tokens, it is not surprising that the difference is small in the shorter length group. As the context length exceeds the optimal chunk size, the baseline model experiences performance degradations with increasing severity. Surprisingly, it is more effective to select a chunk at random than to process the whole sequence, a result that aligns with the AR curves in Figures 1, 2. Lastly, we see that min entropy outperforms all other methods, especially as context length increases.

**Ablating the IDK Filter.** We ablate the IDK Filter on the four Document QA tasks from LongBench_e. Our quantitative findings are shown in Table 6, and qualitative examples are shown in Figure 7 (Appendix). The IDK Filter is highly relevant for the Falcon-Mamba-Inst-7B and Recurrent-Gemma-IT-9B models, especially as context lengths increase. For RWKV6-Finch-7B we see a similar trend, yet it starts a bit later only in the longer length group '8k+'. We hypothesize that the IDK Filter's contribution increases with context length because the number of chunks also increases. With more chunks, the likelihood that one will receive a high confidence score for 'IDK' rises. Lastly, despite not making good use of the IDK Filter, Falcon3-Mamba-Inst-7B still yields good results without it. We believe that with additional fine-tuning the Falcon3 model could further benefit from the

| Method | 0-4K | 4K-8K | 8K+ |
|---|---|---|---|
| Falcon-Mamba-7B-Instruct | 34.21 | 26.94 | 16.86 |
| + OPRM | 35.28 | 30.94 | 27.02 |
| + OPRM + IDK Filter | **37.41** | **34.49** | **36.25** |
| Falcon3-Mamba-7B-Instruct | 35.38 | 26.94 | 26.08 |
| + OPRM | **38.90** | **32.69** | **35.58** |
| + OPRM + IDK Filter | 26.36 | 24.23 | 30.08 |
| Recurrent-Gemma-IT-9B | 27.35 | 23.70 | 16.68 |
| + OPRM | 26.13 | 23.43 | 17.62 |
| + OPRM + IDK Filter | **37.49** | **34.95** | **35.59** |
| RWKV6-Finch-7B | 22.93 | 11.73 | 10.82 |
| + OPRM | **26.22** | 21.58 | 16.19 |
| + OPRM + IDK Filter | 25.07 | **21.62** | **24.88** |

Table 6: **Ablating the IDK Filter.** We test different LLMs on all 4 Document QA tasks (LongBench_e), and average the scores. For most models, the IDK Filter adds a significant boost, especially as context lengths increase.

IDK Filter, yet in this work we decide to remain in a training-free regime. In the qualitative examples, we see that when we apply OPRM without the IDK Filter, the model responds that 'the provided text does not contain any information.' When adding the IDK Filter, the model recovers the correct chunk after the pre-fill phase, and decodes the correct answer.

**Sensitivity to Chunk Size.** We evaluate a Falcon-Mamba-7B-Inst + OPRM model on the LongBench benchmarks using different chunk sizes, $L \in \{1000, 2000, 3000\}$, and compute the variation of the results (Table 7). For most benchmarks the std is only a few percentage points of the mean score, indicating that the method is robust to the choice of $L$.

| Benchmark | HP | Mu | 2Wi | MF | Nar | Qas | GR | QMS | MN | TQA | SAM | TREC | PC | PR | LCC | RB |
|---|---|---|---|---|---|---|---|---|---|---|---|---|---|---|---|---|
| $\sigma / \mu$ | 0.03 | 0.03 | 0.06 | 0.03 | 0.09 | 0.15 | 0.01 | 0.06 | 0.00 | 0.07 | 0.03 | 0.27 | 0.12 | 0.30 | 0.00 | 0.02 |

Table 7: **Chunk size ablation.** We evaluate Falcon-Mamba-Instruct-7B + OPRM using $L \in \{1000, 2000, 3000\}$ on all LongBench benchmarks, and report the std of the results, normalized by their mean. We find that in the majority of the cases OPRM inference is highly robust to different values of $L$. Task name abbreviations can be found in Appendix F.

## 5.4 Efficiency of OPRM

In Table 8 we report the inference time and memory usage of a Falcon3-Mamba-Inst-7B model with and without OPRM inference (pre-fill + decoding of 10 tokens) on a Nvidia RTX A6000 GPU. In this test we use a chunk size of $L = 2000$ tokens. We find that as context length grows, OPRM becomes increasingly faster than the baseline. This is not surprising, as OPRM allows additional parallelization in the sequence dimension (chunking), and reduces complexity, as detailed in Appendix D. Furthermore, we find that OPRM 's memory usage is highly competitive w.r.t the baseline model, despite using much more states. For example, when the context length is 128K tokens, we use 64 chunks, which requires storing 63 more states in the GPU's memory. The relatively small increase in memory usage is due to the fact that the memory occupied by a single state is about three orders of magnitude smaller than the size occupied by the model's weights. This allows us to significantly increase the number of states (chunks) with only a slight increase in total memory usage.

| Model | | 2K | 4k | 8K | 16K | 32K | 64K | 128K |
|---|---|---|---|---|---|---|---|---|
| | | | | | Context Length | | | |
| Time [s] | Falcon3-Mamba-Inst-7B + OPRM | 2.2 | 2.5 | 3.2 | 4.7 | 7.8 | 14.0 | 26.9 |
| | Falcon3-Mamba-Inst-7B | 2.0 | 2.5 | 3.5 | 5.7 | 10.2 | 18.9 | 36.2 |
| Space [GB] | Falcon3-Mamba-Inst-7B + OPRM | 15.3 | 15.6 | 16.3 | 17.8 | 20.6 | 26.2 | 37.3 |
| | Falcon3-Mamba-Inst-7B | 14.9 | 15.3 | 15.9 | 17.2 | 19.9 | 25.2 | 35.7 |

Table 8: **Efficieny of OPRM.** Inference time (seconds) and peak memory usage (GigaBytes) benchmarked on a Nvidia RTX A6000 GPU. For OPRM inference we use $L = 2000$. We find that OPRM outperforms vanilla inference in speed, while adding a surprisingly small memory overhead.

## 6 Limitations

While our overflow-prevention algorithm yields significant gains, it can still be improved. For example, non-trivial overflow-aware cross-chunk aggregation methods would allow for better utilization of global in-context dependencies. Additionally, OPRM is training-free, therefore it relies on the trained model's abilities. E.g., the IDK filter could be better adapted to Falcon3-Mamba-Inst-7B via additional fine-tuning.

## 7 Conclusions

In this paper, we investigate the memory overflow phenomenon in large recurrent models and demonstrate how it limits their long-context performance. To overcome this limitation, we propose OPRM, a training-free overflow-prevention mechanism that operates during inference time. By applying our method, we prevent overflows and achieve significant performance improvements in both real-world and synthetic tasks. Additionally, we find that overflow-prevention methods are highly effective at context extension and let existing recurrent LLMs match or even outperform leading Transformer baselines, while maintaining sub-quadratic efficiency. Lastly, our results raise questions about whether recurrent models genuinely exploit long-range dependencies across multiple chunks, as our single-chunk method leads to stronger performance in a variety of tasks.

*Acknowledgments*

This work was supported by a grant from the Tel Aviv University Center for AI and Data Science (TAD), and was partially supported by the KLA foundation. We would like to thank Han Guo for fruitful discussions and valuable input which helped improve this work.

## 8 Reproducibility Statement

First, we provide the source code used for the key experiments. Second, in appendix A we provide full configurations for all experiments including instructions on how to evaluate the models. We also explicitly state all used datasets, models and hardware, and describe the data processing pipeline and exact metrics used in each experiment.

## 9 Ethics Statement

This work analyzes and improves long-context understanding, which is crucial when deploying LLMs in real-world systems. This improvement is anticipated to have a positive impact on the use of LLMs in society. However, we acknowledge that LLMs can propagate biases. We emphasize the necessity of further research into these biases before our work can be applied reliably beyond the research environment.

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

# A  Implementation Details

All model checkpoints are taken from the Hugging Face Model Hub[1]:

- `tiiuae/falcon-mamba-7b-instruct`
- `tiiuae/Falcon3-Mamba-7B-Instruct`
- `google/recurrentgemma-9b-it`
- `RWKV/v6-Finch-7B-HF`
- `state-spaces/mamba-1.4b`
- `state-spaces/mamba-130m`
- `assafbk/decimamba-130m-niah`
- `assafbk/mamba-130m-niah`

Our code is based on the official Huggingface[2] and Mamba[3] implementations.

## A.1  Zero-Shot Associative Recall

We use contexts of length $N = 1200$. The keys are random 3 letter strings (e.g. 'zqc'), and values are 5 digit numbers (e.g. '38271'). We use the model's tokenizer to tokenize all keys and values, which are, on average, 2-3 tokens long. For padding we use token 29104 ("\n\n\n\n"). During evaluation, for each number of facts $M$ we sample 5 different contexts and report the average performance (for each context we evaluate all $M$ queries). Our padding is evenly spaced.

## A.2  Associative Recall with 2-Layer Models in the Controlled Setup

We use contexts of length $N = 384$. All key, value and padding tokens are one token long. For training we use a blend of information densities: $M \in \{1, 2, 4, 8, 16, 32, 64, 128\}$ facts. Batch creation for training: First, we sample one context for each length $M$. Then we create copies of each context, one for every query in it, and append the query to the end of the context. If $M > 16$ then we generate only 16 copies and sample 16 queries at random. During training we use batch size of 112, and train the model for 18K steps (one epoch). We use a learning rate of 1e-3 and weight decay of 0.1. We train each model with 3 different seeds and average the results. During evaluation, for each number of facts $M$, we average the performance over 100 different contexts (for each context we evaluate all $M$ queries). Our padding is evenly spaced.

## A.3  LongBench and LongBench v2

We apply OPRM inference with $L \in \{1000, 2000, 3000\}$ and select the best scoring chunk size. For padding we use token id 29104 ("\n\n\n\n"). We apply the IDK filter for all models, except for Falcon3-Mamba-Inst-7B. In LongBench, the IDK filter is applied only to Multi-Document QA and Single Document QA tasks. This is because other tasks (summarization, few-shot, code completion, etc.) all have relevant information in all chunks. In the Qasper benchmark, in order to match the correct format, when using the IDK Filter, we replace each predicted "Error" response with "Unanswerable". For the summarization tasks, we found that decoding all chunks in parallel and then concatenating the predictions into a single summary yields better performance. In these cases, we added an additional -Summ to the method name in the table. For the LCC benchmark, we always select the last chunk since the model is required to complete the next line of code. For the Falcon-Mamba models the padding token is 29104 ("\n\n\n\n") and the idk token is 5801 ("Error"), for Recurrent-Gemma the padding token is 0 ("<pad>") and the idk token is 2876 ("Error"), and

---

[1]https://www.huggingface.co/models
[2]https://github.com/huggingface/transformers
[3]https://github.com/state-spaces/mamba

for RWKV the padding token is 261 ("\n\n") and the idk tokens are 33079, 36638 ("Error", " Error"). We use the same prompts used in the LongBench repository. For LongBench: `https://github.com/THUDM/LongBench/blob/main/LongBench/config/dataset2prompt.json`. For LongBench v2 we used the same template: `https://github.com/THUDM/LongBench/blob/main/prompts/0shot.txt`, but specified for the model to output only the letter:

```
Please read the following text and answer the question below.

<text>
$DOC$
</text>

What is the correct answer to this question: $Q$
Choices:
(A) $C_A$
(B) $C_B$
(C) $C_C$
(D) $C_D$

Respond only with the correct letter and do not output anything else. The correct answer is choice "
```

### A.4 Needle in a Haystack

We use the same setup as in Ben-Kish et al. (2025), and use the provided models in the HuggingFace Hub: 'assafbk/mamba-130m-niah' and 'assafbk/decimamba-130m-niah'. We apply OPRM inference on top of the baseline model, and use chunk size $L = 8000$. We average 20 samples per data point.

### A.5 Document Retrieval

We use the same setup as in Ben-Kish et al. (2025). We train each model with data from SQuAD v2 (Rajpurkar et al., 2018), which provides examples in the form of (Query, Document, Answer). The training samples have the following form: $N_{docs} \times <Document>$; $<Answer>$, where $<Document>$ can be either the golden document (which holds the answer to the query) or one of $N_{docs} - 1$ randomly sampled documents. $<Answer>$ holds the id of the golden document. In this setting $N_{docs} = 11$, the order of the documents is random, and the query and respective document id are appended to the beginning of each document. During evaluation the same setting is usedm but the value of $N_{docs}$ is varied between 11 and 240. (between 2,200 tokens to 50,000 tokens). We train the 1.4b models for 400 steps, use a learning rate of 2e-5, gradient clipping of 1, batch size of 64 (used batch accumulation), and AdamW optimizer with weight decay of 0.1. For DeciMamba we use decimation_layer = 11, $L_{base}$=2000 during training and $L_{base}$=5000 during evaluation. We apply OPRM inference on top of the baseline model, and use chunk size $L = 6000$. We train the 130m models for two epochs (1500 steps in each), use a learning rate of 1e-4, gradient clipping of 1, batch size of 64 (used batch accumulation), and AdamW optimizer with weight decay of 0.1. For DeciMamba we use decimation_layer = 12, $L_{base}$=2000 during training and $L_{base}$=4000 during evaluation. We apply OPRM inference on top of the baseline model, and use chunk size $L = 4000$. All results are the average performance over 3 different training seeds, and each data point is evaluated over 100 samples.

## B  Additional Experiments

### B.1  InfiniteBench

InfiniteBench (Zhang et al., 2024) comprises both synthetic and real-world tasks across diverse domains, designed to evaluate a model's ability to understand long-range dependencies within extended contexts exceeding 100k tokens.
Table 9 shows a comparison between Falcon3-Mamba-Inst-7B with and without OPRM, and

additional Attention-based models. Consistent with our earlier findings, OPRM significantly outperforms the baseline, and even beats the equivalently-sized Attention-based model, Mistral-YaRN-7B . Moreover, with OPRM, Falcon3-Mamba is competitive with Attention-based models that have more than x10 parameters such as Claude2 and Kimi-Chat. Lastly, we note that Code.Run is not solved by any model (except GPT4, to some extent). This synthetic task - which requires computing a complex composition of many functions - remains unsolved even by leading proprietary models. The same is true for Math.Calc.

| Model | Kimi-Chat | Claude2 | GPT4 | Mistral-YaRN | Falcon3-Mamba | Falcon3-Mamba + OPRM |
|---|---|---|---|---|---|---|
| Model Type | Transformer | Transformer | Transformer | Transformer | Recurrent | Recurrent |
| # Params | >70B | >70B | >70B | 7B | 7B | 7B |
| Ret.PassKey | 98.1 | 97.8 | 100.0 | 92.7 | 0.0 | **99.8** |
| Ret.Number | 95.4 | 98.1 | 100.0 | 56.6 | 0.0 | **100.0** |
| Ret.KV | 53.6 | 65.4 | 89.0 | 0.0 | 0.0 | **31.3** |
| En.Sum | 17.9 | 14.5 | 14.7 | 9.1 | 20.1 | **22.08** |
| En.QA | 16.5 | 12.0 | 22.2 | 9.6 | 11.0 | **23.2** |
| En.MC | 72.5 | 62.9 | 67.3 | 28.0 | 45.4 | **59.4** |
| En.Dia | 11.5 | 46.5 | 8.5 | 7.5 | 4.0 | **8.0** |
| Code.Dbg | 18.0 | 2.3 | 39.6 | 0.8 | **27.1** | 24.56 |
| Code.Run | 2.0 | 2.5 | 23.3 | **1.3** | 0.0 | 0.0 |
| Math.Calc | 0.0 | 0.0 | 0.0 | **0.0** | **0.0** | **0.0** |
| Math.Find | 12.6 | 32.3 | 60.0 | 17.1 | 26.86 | **33.14** |
| Avg | 36.2 | 39.5 | 47.7 | 20.2 | 12.2 | **36.5** |

Table 9: **InfiniteBench.** Results for Falcon3-Mamba-Inst-7B, with and without OPRM inference, along with Attention-based LLMs. Large proprietary models with a parameter count larger by more than 10 times are added in gray. We find that with OPRM, leading recurrent LLMs, such as Falcon3-Mamba-Inst-7B, can outperform similar sized Transformer-based models, and are comparable to larger proprietary models.

## B.2 Comparison to RAG

RAG methods augment LLMs with information retrieved from a large input source, such as a large document database (Lewis et al., 2020). Since RAG methods can be easily applied to long-context tasks, we compare them to OPRM. As OPRM is training-free, we specifically compare it to two zero-shot RAG methods: DRAGON, which uses a dense retriever designed to generalize in zero-shot settings (Lin et al., 2023), and PRP, a retriever-free approach that ranks chunks using additional forward passes (Qin et al., 2024a). We evaluate these methods across all four Document QA tasks in LongBench_e (HotpotQA, 2WikiMQA, MultiFieldQA, and Qasper), and report the average performance across all benchmarks. For each RAG method, we tested multiple combinations of num_chunks and chunk_size, ensuring a fair comparison by constraining num_chunks x chunk_size = OPRM_chunk_size. As shown in the table below, OPRM outperforms both methods:

| Method | 0-4K | 4K-8K | 8K+ | Avg |
|---|---|---|---|---|
| Falcon-Mamba-Inst-7B | 34.21 | 26.94 | 19.11 | 26.75 |
| + Dragon | 35.85 | 30.05 | 32.74 | 32.88 |
| + PRP | 36.27 | 32.52 | 34.61 | 34.47 |
| + OPRM | **37.41** | **34.49** | **36.25** | **36.05** |

Table 10: **Comparison to RAG Methods.** We compare the above methods across all four Document QA tasks in LongBench_e and report the average score. 0-4K, 4-8K, and 8k+ are the LongBench_e length groups, and Avg is the overall average score. While RAG methods augment the baseline model, we find that OPRM outperforms them, especially as context lengths increase.

Our results are consistent with prior findings: RAG-based methods generally do not outperform long-context LLMs on long-context tasks (Li et al., 2024; Bai et al., 2024; Xu et al., 2024). To the best of our knowledge, there is no definitive explanation - yet, one hypothesis suggests that longer contiguous contexts help the model uncover multi-hop relations, as some reasoning steps depend on access to previously seen information (Xu et al., 2024). In such cases, given the same length budget, selecting longer segments may be more effective

than using multiple smaller chunks: when a chunk is small, its relevance is more likely to be biased toward surface-level similarity with the query. In contrast, longer segments include additional context that can improve the relevance estimation.

Moreover, from an efficiency perspective - the recurrent LLMs used in our work share the same computational complexity as RAG-based solutions. This stands in contrast to the Transformer-based long-context LLMs used in (Li et al., 2024; Bai et al., 2024; Xu et al., 2024), which incur a substantial computational overhead. Lastly, it is also worth noting that PRP's sequential ranking process makes it significantly slower than both OPRM and DRAGON: PRP requires several minutes to process each sample, whereas OPRM and DRAGON take only a few seconds.

To conclude, these findings further underscore the potential of long-context recurrent LLMs when combined with overflow prevention techniques such as OPRM.

### B.3 Needle in a Haystack

Following the setting in Ben-Kish et al. (2025), a Mamba-130m model needs to retrieve a random 5-digit code (needle) hidden at a random location within a concatenation of articles (haystack) sampled from WikiText (Merity et al., 2016). While the base model was trained on context lengths of 2K tokens, during inference we increase the sequence lengths exponentially from 1K to 512K and record the model's performance for a variety of needle depths within the context. Results can be found in Figure 4. We find that OPRM extends the context to sequences that are ×256 longer than those seen during training. This is in contrast to the dedicated method, which is able to extend the context by ×64. The baseline model (Mamba) is able to extend the context by ×8 (not shown).

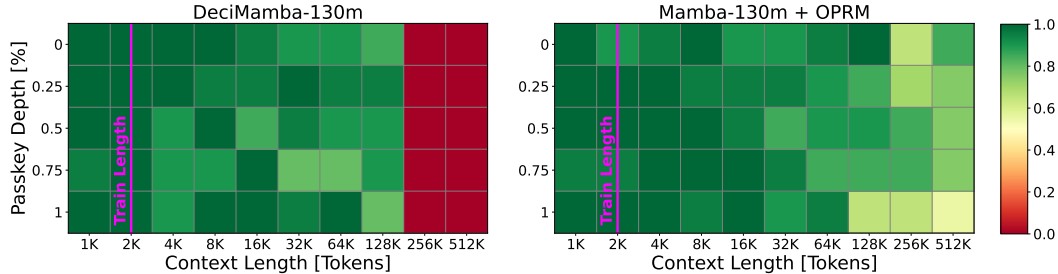

Figure 4: **Context Extension - Needle in a Haystack**. The x-axis is the context length in tokens, and the y-axis is the depth in which the passkey is hidden inside the context. The color indicates the success rate of the needle retrieval. We show that overflow-prevention mechanisms inherently perform context extension, and even beat dedicated context extension methods.

### B.4 Multi-Chunk Ablation

To further motivate our single-chunk approach, we compare OPRM to a multi-chunk strategy. We use the OPRM pipeline to first identify the top-k relevant chunks and then evaluate them together with an additional forward pass. We refer to this method as CC (Combined Chunks). We note that all methods use the same token budget - CC uses smaller chunks with a total length equal to OPRM's single chunk. We evaluated CC's performance across all 4 Document QA tasks in LongBench_e and report the average scores in Table 11.

We see that while CC is beneficial for short contexts (0-4k), it offers no clear advantage over OPRM in longer contexts (4-8k and 8k+). Moreover, it adds computational and algorithmic overheads. One possible explanation is that longer contiguous contexts help the model uncover multi-hop relations (Xu et al., 2024), as some reasoning steps depend on access to previously seen information. In such cases, given the same length budget, selecting longer segments may be better than using multiple smaller chunks: when a chunk is small, its relevance is more likely to be biased toward surface-level similarity with the query. In contrast, longer segments include additional context that can improve the relevance estimation.

We conclude that cross-chunk information fusion warrants further investigation. For ex-

| Method | 0-4K | 4K-8K | 8K+ | Avg |
|---|---|---|---|---|
| Falcon-Mamba-Inst-7B | 34.21 | 26.94 | 19.11 | 26.75 |
| + CC | **40.27** | **35.56** | 35.98 | **37.27** |
| + OPRM | 37.41 | 34.49 | **36.25** | 36.05 |

Table 11: **Multi-Chunk Ablation.** We compare all methods on the 4 Document QA datasets in LongBench_e (HotpotQA, 2WikiMQA, MultiFieldQA, and Qasper) and report the average score for each length group. Falcon-Mamba-Inst-7B is the baseline (vanilla inference). '+' indicates the inference algorithm. The total length of the CC chunks is constrained to equal one OPRM chunk. We experimented with multiple combinations of chunk length and number, and used the best parameters for testing.

ample, a system-oriented approach, such as an overflow-aware variant of Bahdanau Attention (Bahdanau et al., 2016) or hierarchical state processing, may hold significant potential.

## B.5 Sensitivity of Recurrent Memory Capacity to Input Length

We repeat the zero-shot AR experiment (Section 3) with varying sequence lengths and present the results in Figure 5. Our findings indicate that memory capacity is not particularly sensitive to the overall context length. Similar to the original 1,200-token setting, performance starts at around 75% accuracy for 10 key-value pairs (facts), and gradually declines as the number of facts increases, eventually converging toward 0% accuracy. This further demonstrates that recurrent memory overflows represent a fundamentally distinct limitation of recurrent LLMs - one that should be addressed independently of other known limitations, such as length generalization. Lastly, since each fact is 6 tokens long, for a sequence length of L=300 we can only test a maximum of 50 facts and for L=600 a maximum of 100 facts.

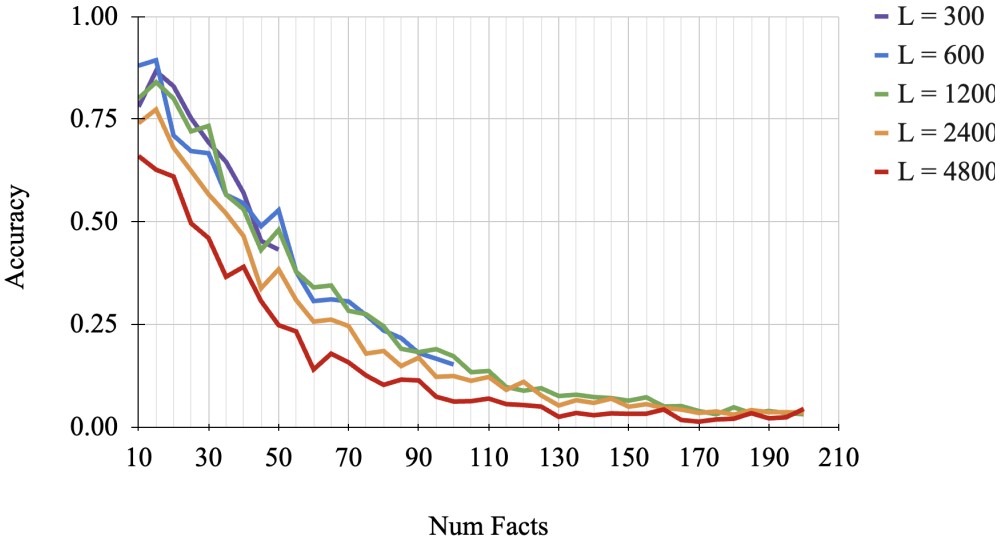

Figure 5: **Sensitivity of recurrent memory capacity to input length**. We repeat the zero-shot AR experiment (Section 3) with varying input sequence lengths $L \in \{300, 600, 1200, 2400, 4800\}$. We find that the model is not sensitive to the input sequence length, but rather to the amount of information within the sequence.

## B.6 Positional Sensitivity to Recurrent Memory Overflows

We analyze the position of successfully retrieved key-value (KV) pairs from the zero-shot AR experiment (Section 3). The results are displayed in Figure 6. Here, each plot shows

a normalized histogram of the locations of successfully retrieved KV pairs (facts). In all measurements we use 10 equally spaced bins to aggregate the samples, and average over 5 different seeds. When the number of facts is small, the distribution appears relatively uniform, which corresponds with the high AR accuracy. Interestingly, as the number of facts increases, we observe a U-shaped recall pattern, resembling the lost-in-the-middle phenomenon reported in (Liu et al., 2024a). We conclude that while recall performance degrades across all positions, the middle portions of the context suffer the most. Since this phenomenon is also observed in Transformer-based LLMs, we suspect that the positional sensitivity may be influenced by properties of the data itself rather than the model's architecture, a finding which warrants further investigation in future work.

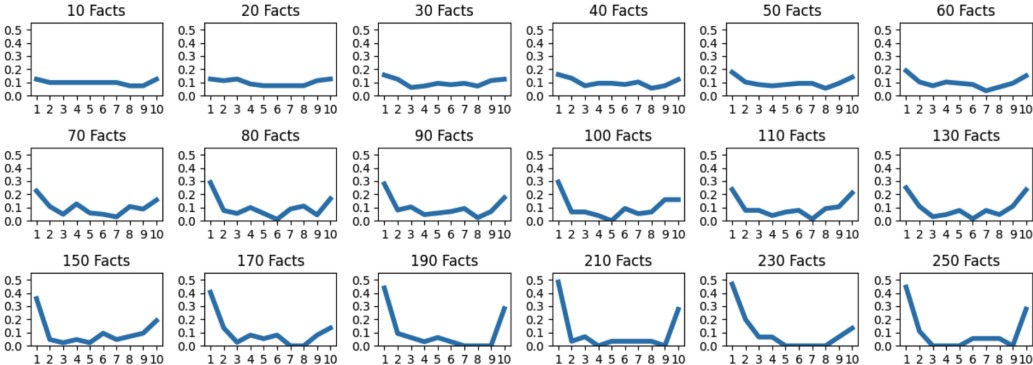

Figure 6: **Positional sensitivity to recurrent memory overflows**. We analyze the position of successfully retrieved key-value (KV) pairs from the zero-shot AR experiment (Section 3). Each plot shows a normalized histogram of the locations of successfully retrieved KV pairs (facts). We find that while recall performance degrades across all positions, the middle portions of the context suffer the most, resembling the "Lost-In-The-Middle" phenomenon observed in Transformer-based LLMs.

### B.7 Comparison to Agentic Long-Context Frameworks

One way to process long contexts is by applying an agentic pipeline that first splits the context into several chunks for LLMs to process and reason over, and then aggregates their intermediate answers using additional LLM calls to produce the final output. These methods contrast with inference methods like OPRM, which process the whole context in a single forward pass. We compare OPRM to LLMxMapReduce, an agentic framework for long context processing (Zhou et al., 2024). We apply LLMxMapReduce to Falcon3-Mamba-Inst-7B, and find that while it produces reasonable answers for some queries, it often fails at following the full set of instructions at different stages of the pipeline. Notably, in their paper, LLMxMapReduce use Transformer models with a minimum size of 70 billion parameters, whereas Falcon3-Mamba-Inst-7B, at 7 billion parameters, is currently the largest available recurrent LLM. We believe this performance gap is largely due to the significant difference in model scale, and conclude that it will be beneficial to evaluate LLMxMapReduce on larger-scale recurrent LLMs, once they become available.

Most importantly, we report that OPRM is over an order of magnitude faster than LLMxMapReduce: while LLMxMapReduce requires 225.2 seconds per query, OPRM completes a query in 8.6 seconds - a total speedup of x26 times.

## C Motivation and Design Principles

Our method is grounded in several key design principles, including the role of locality and compression in NLP, as well as a speculative processing strategy. In this section, we elaborate on these choices and explain how they guide the design of OPRM.

Beyond reducing the context size from $BL$ to $L$ to prevent memory overflows, OPRM is based on the following key ideas:

**Locality.** Each prompt $X_i$ is processed using an inner-chunk modeling approach. This design choice is based on the assumption that natural language exhibits strong local structural properties, where semantic dependencies between distant text segments are inherently limited. By leveraging this locality, the model can efficiently process smaller chunks while preserving the most relevant contextual information. While we acknowledge that decoding based on information from a single chunk cannot capture all global dependencies in the context, we demonstrate through extensive experiments that this approach is highly effective, and leave more sophisticated aggregation methods for future works.

**Recurrent LLMs as Compression and Decompression Models.** By preventing memory overflows, recurrent LLMs can be interpreted as strong compression and decompression mechanisms. We leverage this property by compressing all chunks during processing while decoding only the most relevant one.

**Speculative Approach.** Our approach can be interpreted as a speculative pre-fill strategy, analogous to branch prediction in CPU architectures (Smith, 1998) or speculative decoding for accelerating auto-regressive LLMs (Leviathan et al., 2023; Xia et al., 2024). Specifically, each prompt is processed under the assumption that the relevant context is included in the prompt. Furthermore, we assume that the model's output probability distribution $Pr(\cdot \mid X_i)$ has the necessary information to determine whether the prompt $X_i$ contains the relevant chunk $C_i$, which is essential for generating the answer to Q.

## D   Advantages of OPRM

Beyond its ability to handle long contexts, OPRM offers several unique advantages:

**Efficient Dynamic memory.**   Unlike the fixed-size memory of recurrent LLMs such as Mamba, OPRM's memory capacity increases linearly with context length during prefill (via per-chunk states) while remaining fixed during decoding. This mechanism is designed to mitigate memory overflows during prefill while preserving the most important property that makes recurrent LLMs efficient—constant time and space complexity that is independent of context length during auto-regressive decoding.

**Efficient computation.**   Assuming that the lengthy part of the prompt $X$ is the context $C$ ($|C| \gg |P|, |S|$), OPRM offers two key efficiency advantages over other LLMs: (i) OPRM processes each chunk independently and thus reduces complexity. As most recurrent LLMs employ FFT-based methods or work-efficient parallel scan algorithms during prefill, both leading to an $O(Lb \log Lb)$ aggregated time-complexity for processing a sequence of length $Lb$, OPRM improves efficiency by reducing the prefill time complexity to $O(bL \log L)$. Additionally, (ii) in many cases, the context C and prefix P are known in advance, while the query (suffix) is provided in real time. OPRM can leverage this by pre-computing the recurrent states for each chunk of the context $[P, C_i]$. At inference time, instead of recomputing from scratch, the model initializes with these precomputed states, allowing efficient real-time query processing and significantly reducing computational overhead, resulting in a prefill complexity of $O(b|S|)$ via sequential prefill, which isn't dependent on the context length $|C|$. Note that such an optimization will not be as effective for Transformers, as they do not compress the previously processed inputs.

**Control of the memory-recall tradeoff.**   Our method has a single hyperparameter, the chunk size $L$, which controls the memory-recall tradeoff during prefill. As $L$ increases, the likelihood of memory overflow increases, but the memory capacity (the number of states) also grows as well. Thus, this hyperparameter provides a simple way to balance memory constraints and recall performance based on the task.

**Compatibility with RAG based settings.**   Although our method can be applied to any long-text scenario, as the Prefix-Context-Suffix structure is a general framework, it is also

well-suited for query-based RAG (Lewis et al., 2020). This setting focuses on dynamically retrieving relevant context based on the query before feeding it into the LLM. Our approach is particularly efficient in this setting, as it processes the query with the selected chunk $C_j$ in a single pass performed when computing the state $S_j$. This makes it a simple yet effective drop-in component for various tasks and real-world applications. While our approach shares similarities with RAG methods, we emphasize that our primary focus is not on direct comparisons with RAG benchmarks. Instead, our goal is to enhance recurrent LLMs' long-context capabilities. Furthermore, we find that our method outperforms several RAG methods, as shown and discussed in Section B.2.

| Benchmark | Method | 0-4K | 4K-8K | 8K+ | LB |
|---|---|---|---|---|---|
| HotPotQa (2 hops) | Baseline | 27.97 | 21.57 | 17.21 | 22.17 |
| | + OPRM | **38.68** | **34.37** | **36.07** | **35.09** |
| | *Improvement* | *38.3%* | *59.3%* | *109.7%* | *58.3%* |
| MuSiQue ($\leq$ 4 hops) | Baseline | N/A | N/A | N/A | 8.37 |
| | + OPRM | N/A | N/A | N/A | **18.4** |
| | *Improvement* | *N/A* | *N/A* | *N/A* | *119.8%* |
| 2WikiMQA ($\leq$ 5 hops) | Baseline | 25.26 | 25.33 | 16.61 | 21.39 |
| | + OPRM | **30.37** | **28.88** | **27.01** | **25.08** |
| | *Improvement* | *20.2%* | *14.0%* | *62.7%* | *17.2%* |

Table 12: **MultiHop Reasoning.** We test SoTA recurrent LLMs with and without OPRM on all multi-hop benchmarks in LongBench, and report the average score over all models: Falcon3-Mamba-Inst-7B, Falcon-Mamba-Inst-7B, RecurrentGemma-IT-9B, and RWKV6-Finch-7B. *Improvement* shows the relative improvement in %. 0-4K, 4-8K, and 8k+ are the LongBench_e length groups, LB is the LongBench score. OPRM significantly improves multi-hop reasoning capabilities, sometimes by even more than 100%. The data is taken from Tables 13, 14, 15, 16 in the Appendix.

# E    Additional Related Work

## E.1    Context Extension Methods

### E.1.1    Sub-Quadratic Models

Ben-Kish et al. (2025) identify that Mamba-based models overfit to the lengths that they were trained on, a behavior that leads to a complete collapse during length generalization. To overcome this limitation, they propose DeciMamba, a global token filtering mechanism, which achieves significant length generalization by discarding tokens that are considered unimportant by the S6 layer. Ye et al. (2025) builds upon this finding and propose implementing a similar mechanism at the channel level, allowing finer pooling. Azizi et al. (2025) additionally builds upon this finding and adds learnable scaling factors that keep the values of the state-space parameters within a valid range during length generalization. While these methods extend the amount of tokens that the model can process without collapsing, they do not guarantee that the model will be able to store all relevant information in the context, as shown in Sec. 3, hence prone to memory overflows as well.

### E.1.2    Transformers

Several methods were proposed to enhance the effective context length of Transformers and improve their extrapolation over longer sequences. Press et al. (2021) demonstrated that models built on top of original sinusoidal, rotary (Su et al., 2024), and T5 bias (Raffel et al., 2020) positional encoding have poor length generalization. It proposed to mitigate this issue by incorporating distance-based linear biases into the attention matrix, thus promoting locality. Kazemnejad et al. (2024) showed that Transformers without positional encoding (NoPE) exhibit better length extrapolation capabilities in downstream tasks. Another

recent direction involves architectural modifications to pre-trained models followed by short fine-tuning. It includes LongLora (Chen et al., 2023), which proposes shifted sparse attention, and Landmark Attention (Mohtashami & Jaggi, 2023), which applies attention in chunks and inserts global unique tokens into the input sequences between those chunks. Although context extension for Transformer models is not trivial, it does not have to deal with memory overflows, since the memory size (KV-Cache) grows with the amount of tokens in the sequence. This is in contrast to sub-quadratic models, which have a fixed memory capacity, hence require additional care.

## E.2 Retrieval Augmented Generation (RAG)

Augmenting LLMs with retrieval capabilities has become a well-established strategy for improving factual accuracy (Nakano et al., 2022), enhancing downstream task performance (Guu et al., 2020; Izacard et al., 2022; Lewis et al., 2020), and improving long-context capabilities (Li et al., 2024; Xia et al., 2024). Typically, this is done by augmenting the model with a separate retriever model which is trained on downstream data (Karpukhin et al., 2020). However, it was shown that it is challenging to deploy such retrievers in real-world scenarios where training data is scarce (Thakur et al., 2021). To mitigate this, recent works train RAG systems for increased generalization, allowing them to operate in zero-shot scenarios (Lin et al., 2023; Formal et al., 2021). While these methods improve performance, they have not demonstrated superiority over long-context models (Li et al., 2024; Bai et al., 2024; Xia et al., 2024). Our work is consistent with these findings - we show that by preventing memory overflows with OPRM, long-context recurrent LLMs achieve superior results with respect to RAG approaches (Section B.2).
Other works replace the retriever with additional LLM-based processing, using the model itself to rank input segments (Qin et al., 2024a). While the LLMs have better generalization capabilities, the additional forward passes are costly and result in significant slowdowns with respect to a typical retriever model. In contrast to these approaches, OPRM does not follow the Retrieve-Prefill-Decode paradigm of RAG-based methods. Instead, it computes the whole context in a single forward pass (Prefill-Decode), like any other LLM. This allows both effectiveness and efficiency, as shown in Section B.2.

## E.3 Mamba - Full Definition

Given an input sequence $U = (u_1, u_2, \ldots, u_L) \in \mathbb{R}^{L \times d}$ of length $L$ such that $u_i \in \mathbb{R}^d$, a Mamba block with $d$ channels is built on top of the S6 layer via the following formula:

$$G = \sigma(\text{Linear}(U)), \quad X = \text{Conv1D}(\text{Linear}(U)),$$
$$Y = S6(X), \qquad O = Y \otimes G \tag{1}$$

where $G$ represents the gate branch, $\otimes$ is elementwise multiplication, $\sigma$ is the SILU activation, Linear and Conv1D are standard linear projection and 1-dimensional convolution layers. The S6 layer is based on a time-variant SSM, which can be elaborated by the following recurrent rule:

$$h_t = \bar{A}_t h_{t-1} + \bar{B}_t x_t, \quad y_t = C_t h_t \tag{2}$$

where $\bar{A}_t \in \mathbb{R}^{d_{state} \times d_{state}}$, $\bar{B}_t \in \mathbb{R}^{d_{state} \times 1}$, and $C_t \in \mathbb{R}^{1 \times d_{state}}$ are the system, input, and output discrete time-variant matrices, respectively.
Lastly, $X = (x_1, x_2, \ldots, x_L)$ is the input sequence of a representative channel. S6 conditions the discrete time-variant matrices based on the input as follows:

$$\Delta_t = Sft(S_\Delta X_t), B_t = S_B X_t, C_t = (S_C X_t)^T$$
$$\bar{A}_t = \exp(A\Delta_t), \quad \bar{B}_t = B_t \Delta_t \tag{3}$$

such that $\Delta_t$ is the discretization step, $Sft$ represents the softplus function, and $S_\Delta, S_B, S_C$ are linear projection layers. As each Mamba channel has a state of size $d_{state}$, the hidden state size is defined as $d \times d_{state}$.

# F   LongBench Task Name Abbreviations.

HP, Mu, 2Wi, MF, Nar, Qas, GR, QMS, MN, TQA, SAM, TREC, PC, PR, LCC, and RB stand for HotPotQA, Musique, 2WikiMQA, MultiFieldQA, NarrativeQA, Qasper, GovReport, QMSSum, MultiNews, TriviaQA, SAMSum, TREC, Passage Count, Passage Retrieval en, LCC, and RepoBench -p, respectively.

| Type (Metric) | Benchmark | Avg Len | Model | Benchmark Type | | | |
|---|---|---|---|---|---|---|---|
| | | | | 0-4k | 4-8k | 8k+ | LB |
| MD-QA (F1) | 2wikimqa | 4887 | Falcon-Mamba-Inst-7B | 28.67 | **31.02** | 13.75 | **23.8** |
| | | | Falcon-Mamba-Inst-7B + OPRM | **31.05** | 25.26 | **28.18** | 22.83 |
| MD-QA (F1) | Hotpotqa | 9151 | Falcon-Mamba-Inst-7B | 36.35 | 21.18 | 18.4 | 27.39 |
| | | | Falcon-Mamba-Inst-7B + OPRM | **37.63** | **40.22** | **41.03** | **36.39** |
| MD-QA (F1) | Musique | 11214 | Falcon-Mamba-Inst-7B | N/A | N/A | N/A | 8.53 |
| | | | Falcon-Mamba-Inst-7B + OPRM | N/A | N/A | N/A | **20.52** |
| SD-QA (F1) | Narrative QA | 18409 | Falcon-Mamba-Inst-7B | N/A | N/A | N/A | 7.8 |
| | | | Falcon-Mamba-Inst-7B + OPRM | N/A | N/A | N/A | **20.55** |
| SD-QA (F1) | Qasper | 3619 | Falcon-Mamba-Inst-7B | 29.35 | 26.64 | 14.8 | 30.2 |
| | | | Falcon-Mamba-Inst-7B + OPRM | **35.74** | **35.15** | **33.79** | **34.7** |
| SD-QA (F1) | Multifield QA | 4559 | Falcon-Mamba-Inst-7B | 42.48 | 28.93 | 20.48 | 33.89 |
| | | | Falcon-Mamba-Inst-7B + OPRM | **45.21** | **37.33** | **41.98** | **41.88** |
| Summ (Rouge-L) | GovReport | 8734 | Falcon-Mamba-Inst-7B | 29.65 | 24.23 | 18.81 | 22.04 |
| | | | Falcon-Mamba-Inst-7B + OPRM-Summ | **35.56** | **36.61** | **35.74** | **35.88** |
| Summ (Rouge-L) | QMSum | 10614 | Falcon-Mamba-Inst-7B | N/A | N/A | N/A | 19.15 |
| | | | Falcon-Mamba-Inst-7B + OPRM-Summ | N/A | N/A | N/A | **19.79** |
| Summ (Rouge-L) | MultiNews | 2113 | Falcon-Mamba-Inst-7B | 26.75 | 21.42 | 16.61 | 25.8 |
| | | | Falcon-Mamba-Inst-7B + OPRM-Summ | **27.35** | **25.31** | **21.04** | **27.14** |
| Few-Shot (F1) | TriviaQA | 8209 | Falcon-Mamba-Inst-7B | **73.56** | **78.47** | 69.68 | 69.81 |
| | | | Falcon-Mamba-Inst-7B + OPRM | 71.01 | 78.41 | **73.05** | **72.74** |
| Few-Shot (Rouge-L) | SAMSum | 6258 | Falcon-Mamba-Inst-7B | **38.84** | **37.4** | 31.02 | 37.62 |
| | | | Falcon-Mamba-Inst-7B + OPRM | 38.22 | 35.12 | **37.85** | **40.72** |
| Few-Shot (Acc.) | TREC | 5177 | Falcon-Mamba-Inst-7B | 17.0 | 11.0 | 1.0 | 10.5 |
| | | | Falcon-Mamba-Inst-7B + OPRM | **45.0** | **42.0** | **48.0** | **43.5** |
| Code (Edit Sim) | LCC | 1235 | Falcon-Mamba-Inst-7B | **44.16** | **34.4** | **23.41** | **40.44** |
| | | | Falcon-Mamba-Inst-7B + OPRM | **44.16** | **34.4** | **23.41** | **40.44** |
| Code (Edit Sim) | RepoBench-p | 4206 | Falcon-Mamba-Inst-7B | **30.81** | 26.58 | 20.07 | 32.0 |
| | | | Falcon-Mamba-Inst-7B + OPRM | 28.62 | **29.35** | **33.05** | **33.08** |
| Syn (Acc.) | Passage Count | 11141 | Falcon-Mamba-Inst-7B | 5.0 | 4.0 | **7.0** | 2.0 |
| | | | Falcon-Mamba-Inst-7B + OPRM | **12.0** | **6.0** | 6.0 | **5.0** |
| Syn (Acc.) | Passage Ret (en) | 9289 | Falcon-Mamba-Inst-7B | 10.0 | 4.0 | 8.0 | 4.5 |
| | | | Falcon-Mamba-Inst-7B + OPRM | **30.0** | **19.0** | **15.0** | **12.5** |

Table 13: **LongBench - full results for Falcon-Mamba-Instruct-7B.** We show the results with and without OPRM Inference for LongBench (LB) and LongBench-E (0-4k, 4-8k, 8k+). MD-QA, SD-QA, Summ, Syn, Passage Ret (en) stand for MultiDocument-QA, Single Document-QA, Summarization, Synthetic, Passage Retrieval (english). Avg Len is the Average Length in words. Falcon-Mamba-Inst-7B + OPRM-Summ uses the summarization technique detailed in Appendix A.3.

| Type (Metric) | Benchmark | Avg Len | Model | Benchmark Type | | | |
|---|---|---|---|---|---|---|---|
| | | | | 0-4k | 4-8k | 8k+ | LB |
| MD-QA (F1) | 2wikimqa | 4887 | Falcon3-Mamba-Inst-7B | 36.05 | 32.49 | 26.11 | 28.03 |
| | | | Falcon3-Mamba-Inst-7B + OPRM | **42.20** | **34.68** | **34.34** | **33.61** |
| MD-QA (F1) | Hotpotqa | 9151 | Falcon3-Mamba-Inst-7B | 38.50 | 29.51 | 29.53 | 30.88 |
| | | | Falcon3-Mamba-Inst-7B + OPRM | **48.03** | **42.19** | **41.79** | **45.17** |
| MD-QA (F1) | Musique | 11214 | Falcon3-Mamba-Inst-7B | N/A | N/A | N/A | 12.17 |
| | | | Falcon3-Mamba-Inst-7B + OPRM | N/A | N/A | N/A | **22.79** |
| SD-QA (F1) | Narrative QA | 18409 | Falcon3-Mamba-Inst-7B | N/A | N/A | N/A | 15.48 |
| | | | Falcon3-Mamba-Inst-7B + OPRM | N/A | N/A | N/A | **19.30** |
| SD-QA (F1) | Qasper | 3619 | Falcon3-Mamba-Inst-7B | **35.09** | **21.46** | 31.13 | **28.89** |
| | | | Falcon3-Mamba-Inst-7B + OPRM | 26.38 | 20.43 | **32.32** | 25.45 |
| SD-QA (F1) | Multifield QA | 4559 | Falcon3-Mamba-Inst-7B | 31.88 | 24.30 | 17.56 | 27.25 |
| | | | Falcon3-Mamba-Inst-7B + OPRM | **38.97** | **33.46** | **33.87** | **35.99** |
| Summ (Rouge-L) | GovReport | 8734 | Falcon3-Mamba-Inst-7B | 32.08 | 29.55 | 27.17 | 28.23 |
| | | | Falcon3-Mamba-Inst-7B + OPRM-Summ | **33.79** | **35.25** | **32.24** | **33.5** |
| Summ (Rouge-L) | QMSum | 10614 | Falcon3-Mamba-Inst-7B | N/A | N/A | N/A | **19.56** |
| | | | Falcon3-Mamba-Inst-7B + OPRM-Summ | N/A | N/A | N/A | 18.13 |
| Summ (Rouge-L) | MultiNews | 2113 | Falcon3-Mamba-Inst-7B | 24.8 | 21.31 | **19.86** | 24.95 |
| | | | Falcon3-Mamba-Inst-7B + OPRM-Summ | **25.14** | **23.06** | 18.02 | **25.09** |
| Few-Shot (F1) | TriviaQA | 8209 | Falcon3-Mamba-Inst-7B | 79.36 | **85.39** | **78.96** | **78.84** |
| | | | Falcon3-Mamba-Inst-7B + OPRM | **79.86** | 80.66 | 78.17 | 78.2 |
| Few-Shot (Rouge-L) | SAMSum | 6258 | Falcon3-Mamba-Inst-7B | 31.35 | **29.16** | 31.55 | 31.83 |
| | | | Falcon3-Mamba-Inst-7B + OPRM | **32.29** | 29.14 | **32.98** | **32.45** |
| Few-Shot (Acc.) | TREC | 5177 | Falcon3-Mamba-Inst-7B | 27.00 | **58.00** | **55.00** | 42.00 |
| | | | Falcon3-Mamba-Inst-7B + OPRM | **44.00** | 46.00 | 49.00 | **48.50** |
| Code (Edit Sim) | LCC | 1235 | Falcon3-Mamba-Inst-7B | **36.85** | **38.78** | 29.27 | **39.51** |
| | | | Falcon3-Mamba-Inst-7B + OPRM | 36.56 | 38.44 | **29.40** | **39.51** |
| Code (Edit Sim) | RepoBench-p | 4206 | Falcon3-Mamba-Inst-7B | 32.49 | 32.68 | 35.78 | 36.71 |
| | | | Falcon3-Mamba-Inst-7B + OPRM | **37.33** | **35.86** | **38.43** | **40.35** |
| Syn (Acc.) | Passage Count | 11141 | Falcon3-Mamba-Inst-7B | 9.00 | **7.00** | 4.17 | 1.50 |
| | | | Falcon3-Mamba-Inst-7B + OPRM | **10.00** | **7.00** | **7.00** | **5.50** |
| Syn (Acc.) | Passage Ret (en) | 9289 | Falcon3-Mamba-Inst-7B | 12.00 | 3.00 | 9.00 | 6.50 |
| | | | Falcon3-Mamba-Inst-7B + OPRM | **17.00** | **12.00** | **12.00** | **12.00** |

Table 14: **LongBench - full results for Falcon3-Mamba-Instruct-7B.** We show the results with and without OPRM inference for LongBench (LB) and LongBench-E (0-4k, 4-8k, 8k+). MD-QA, SD-QA, Summ, Syn, Passage Ret (en) stand for MultiDocument-QA, Single Document-QA, Summarization, Synthetic, Passage Retrieval (english). Avg Len is the Average Length in words. Falcon3-Mamba-Inst-7B + OPRM-Summ uses the summarization technique detailed in Appendix A.3.

Question: Josephine de Reszke made her debut in what grand opera?

GT Answer: Hamlet

Recurrent-Gemma-IT-9B: "*The provided text does not contain any information regarding Josephine de Reszke's debut in any grand opera.*"

Recurrent-Gemma-IT-9B + OPRM: "*The provided text does not contain any information regarding Josephine de Reszke's debut in a grand opera.*"

Recurrent-Gemma-IT-9B + OPRM + IDK Filter: "*Hamlet*"

Context Length: 10673

Question: William Ulmer Brewery is located in a neighborhood that is policed by who?

GT Answer: NYPD's 83rd Precinct

Recurrent-Gemma-IT-9B: "*The William Ulmer Brewery is located in a neighborhood that is policed by the New York City Police.*"

Recurrent-Gemma-IT-9B + OPRM: "*The provided text does not contain any information regarding the police jurisdiction of the neighborhood where the William Ulmer...*"

Recurrent-Gemma-IT-9B + OPRM + IDK Filter: "*NYPD's 83rd Precinct*"

Context Length: 10365

Question: Which town near the county border with North Yorkshire was this Lancashire mill (closed in 1979 and demolished) located?

GT Answer: Barnoldswick

Recurrent-Gemma-IT-9B: "*The town near the county border with North Yorkshire where this Lancashire mill (closed in 197...*"

Recurrent-Gemma-IT-9B + OPRM: "*The provided text does not contain any information regarding a Lancashire mill or its location, so I cannot answer.*"

Recurrent-Gemma-IT-9B + OPRM + IDK Filter: "*Barnoldswick*"

Context Length: 8159

Figure 7: **IDK Filter ablation - qualitative example** We provide 3 samples from the Hot-PotQA benchmark (LongBench_e). In each one of the samples we present the question, ground truth answer, and the response of three Recurrent-Gemma-IT-9B models: the baseline, + OPRM, and + OPRM + IDK Filter. Each response is colored according to its correctness (red - incorrect, green - correct). The model with OPRM does have the ability to answer the long-context questions, yet without the IDK filter the wrong chunk is selected, leading the model to an 'i don't know' response. When applying the IDK filter, the model provides the exact answer.

| Type (Metric) | Benchmark | Avg Len | Model | Benchmark Type | | | |
|---|---|---|---|---|---|---|---|
| | | | | 0-4k | 4-8k | 8k+ | LB |
| MD-QA (F1) | 2wikimqa | 4887 | Recurrent-Gemma-IT-9B | 25.50 | 18.95 | 14.42 | 24.00 |
| | | | Recurrent-Gemma-IT-9B + OPRM | **31.36** | **36.81** | **28.12** | **31.71** |
| MD-QA (F1) | Hotpotqa | 9151 | Recurrent-Gemma-IT-9B | 27.65 | 18.27 | 12.16 | 21.94 |
| | | | Recurrent-Gemma-IT-9B + OPRM | **46.38** | **35.3** | **43.86** | **37.99** |
| MD-QA (F1) | Musique | 11214 | Recurrent-Gemma-IT-9B | N/A | N/A | N/A | 9.47 |
| | | | Recurrent-Gemma-IT-9B + OPRM | N/A | N/A | N/A | **19.11** |
| SD-QA (F1) | Narrative QA | 18409 | Recurrent-Gemma-IT-9B | N/A | N/A | N/A | 15.11 |
| | | | Recurrent-Gemma-IT-9B + OPRM | N/A | N/A | N/A | **15.79** |
| SD-QA (F1) | Qasper | 3619 | Recurrent-Gemma-IT-9B | 21.52 | 27.29 | 14.82 | 26.14 |
| | | | Recurrent-Gemma-IT-9B + OPRM | **29.24** | **34.49** | **30.16** | **35.54** |
| SD-QA (F1) | Multifield QA | 4559 | Recurrent-Gemma-IT-9B | 34.74 | 30.27 | 25.31 | 31.84 |
| | | | Recurrent-Gemma-IT-9B + OPRM | **42.99** | **33.18** | **40.2** | **38.17** |
| Summ (Rouge-L) | GovReport | 8734 | Recurrent-Gemma-IT-9B | 26.97 | 25.8 | 24.57 | 24.79 |
| | | | Recurrent-Gemma-IT-9B + OPRM-Summ | **33.86** | **36.26** | **34.42** | **34.83** |
| Summ (Rouge-L) | QMSum | 10614 | Recurrent-Gemma-IT-9B | N/A | N/A | N/A | 17.82 |
| | | | Recurrent-Gemma-IT-9B + OPRM-Summ | N/A | N/A | N/A | **19.25** |
| Summ (Rouge-L) | MultiNews | 2113 | Recurrent-Gemma-IT-9B | 23.95 | 19.55 | **18.42** | 23.15 |
| | | | Recurrent-Gemma-IT-9B + OPRM-Summ | **25.13** | **23.46** | 18.22 | **25.27** |
| Few-Shot (F1) | TriviaQA | 8209 | Recurrent-Gemma-IT-9B | 17.54 | 16.85 | 28.77 | 22.06 |
| | | | Recurrent-Gemma-IT-9B + OPRM | **80.91** | **77.97** | **85.36** | **83.85** |
| Few-Shot (Rouge-L) | SAMSum | 6258 | Recurrent-Gemma-IT-9B | 5.58 | 5.05 | 6.36 | 4.50 |
| | | | Recurrent-Gemma-IT-9B + OPRM | **7.22** | **7.50** | **8.92** | **7.83** |
| Few-Shot (Acc.) | TREC | 5177 | Recurrent-Gemma-IT-9B | 0.00 | 0.00 | 0.00 | 0.00 |
| | | | Recurrent-Gemma-IT-9B + OPRM | **9.00** | **16.00** | **7.00** | **11.00** |
| Code (Edit Sim) | LCC | 1235 | Recurrent-Gemma-IT-9B | 37.21 | 38.79 | 40.89 | 41.58 |
| | | | Recurrent-Gemma-IT-9B + OPRM | **46.90** | **50.42** | **42.73** | **46.96** |
| Code (Edit Sim) | RepoBench-p | 4206 | Recurrent-Gemma-IT-9B | 29.25 | 30.94 | **33.16** | 33.57 |
| | | | Recurrent-Gemma-IT-9B + OPRM | **37.99** | **35.54** | 32.30 | **39.86** |
| Syn (Acc.) | Passage Count | 11141 | Recurrent-Gemma-IT-9B | 12.00 | **5.00** | **7.00** | 3.50 |
| | | | Recurrent-Gemma-IT-9B + OPRM | **15.00** | **5.00** | 4.00 | **4.50** |
| Syn (Acc.) | Passage Ret (en) | 9289 | Recurrent-Gemma-IT-9B | 7.20 | 4.20 | **7.00** | **5.52** |
| | | | Recurrent-Gemma-IT-9B + OPRM | **9.00** | **5.00** | 7.00 | 3.00 |

Table 15: **LongBench - full results for Recurrent-Gemma-IT-9B.** We show the results with and without OPRM inference for LongBench (LB) and LongBench-E (0-4k, 4-8k, 8k+). MD-QA, SD-QA, Summ, Syn, Passage Ret (en) stand for MultiDocument-QA, Single Document-QA, Summarization, Synthetic, Passage Retrieval (english). Avg Len is the Average Length in words. Recurrent-Gemma-IT-9B + OPRM-Summ uses the summarization technique detailed in Appendix A.3.

| Type (Metric) | Benchmark | Avg Len | Model | Benchmark Type | | | |
|---|---|---|---|---|---|---|---|
| | | | | 0-4k | 4-8k | 8k+ | LB |
| MD-QA (F1) | 2wikimqa | 4887 | RWKV6-Finch-7B | 10.83 | 18.86 | 12.14 | 9.74 |
| | | | RWKV6-Finch-7B + OPRM | **16.85** | **18.75** | **17.41** | **12.17** |
| MD-QA (F1) | Hotpotqa | 9151 | RWKV6-Finch-7B | 9.37 | 17.33 | 8.73 | 8.47 |
| | | | RWKV6-Finch-7B + OPRM | **22.67** | **19.76** | **17.61** | **20.79** |
| MD-QA (F1) | Musique | 11214 | RWKV6-Finch-7B | N/A | N/A | N/A | 3.31 |
| | | | RWKV6-Finch-7B + OPRM | N/A | N/A | N/A | **11.18** |
| SD-QA (F1) | Narrative QA | 18409 | RWKV6-Finch-7B | N/A | N/A | N/A | 7.31 |
| | | | RWKV6-Finch-7B + OPRM | N/A | N/A | N/A | **10.48** |
| SD-QA (F1) | Qasper | 3619 | RWKV6-Finch-7B | 18.03 | 19.32 | 11.01 | 15.67 |
| | | | RWKV6-Finch-7B + OPRM | **18.88** | **19.83** | **19.12** | **23.53** |
| SD-QA (F1) | Multifield QA | 4559 | RWKV6-Finch-7B | 24.18 | 36.22 | 15.02 | 9.39 |
| | | | RWKV6-Finch-7B + OPRM | **37.55** | **41.95** | **32.32** | **43.03** |
| Summ (Rouge-L) | GovReport | 8734 | RWKV6-Finch-7B | 14.39 | 26.14 | 14.69 | 9.34 |
| | | | RWKV6-Finch-7B + OPRM-Summ | **23.46** | **27.62** | **23.83** | **20.89** |
| Summ (Rouge-L) | QMSum | 10614 | RWKV6-Finch-7B | N/A | N/A | N/A | 12.29 |
| | | | RWKV6-Finch-7B + OPRM-Summ | N/A | N/A | N/A | **22.65** |
| Summ (Rouge-L) | MultiNews | 2113 | RWKV6-Finch-7B | 24.56 | **26.4** | 16.83 | 11.23 |
| | | | RWKV6-Finch-7B + OPRM-Summ | **24.62** | 25.1 | **20.6** | **17.74** |
| Few-Shot (F1) | TriviaQA | 8209 | RWKV6-Finch-7B | 55.65 | 70.63 | 34.81 | 56.88 |
| | | | RWKV6-Finch-7B + OPRM | **69.28** | **72.46** | **64.06** | **70.14** |
| Few-Shot (Rouge-L) | SAMSum | 6258 | RWKV6-Finch-7B | 16.05 | **23.85** | 8.80 | 8.88 |
| | | | RWKV6-Finch-7B + OPRM | **21.89** | 23.28 | **20.55** | **21.61** |
| Few-Shot (Acc.) | TREC | 5177 | RWKV6-Finch-7B | 10.5 | 32.00 | 1.00 | 1.00 |
| | | | RWKV6-Finch-7B + OPRM | **50.50** | **38.00** | **46.00** | **51.00** |
| Code (Edit Sim) | LCC | 1235 | RWKV6-Finch-7B | 25.91 | 28.09 | 19.29 | 20.27 |
| | | | RWKV6-Finch-7B + OPRM | **29.55** | **30.55** | **29.17** | **28.96** |
| Code (Edit Sim) | RepoBench-p | 4206 | RWKV6-Finch-7B | 18.01 | 23.12 | 16.52 | 16.25 |
| | | | RWKV6-Finch-7B + OPRM | **22.68** | **23.33** | **22.63** | **21.25** |
| Syn (Acc.) | Passage Count | 11141 | RWKV6-Finch-7B | 3.00 | 6.00 | 1.00 | 0.00 |
| | | | RWKV6-Finch-7B + OPRM | **6.10** | **9.00** | **6.00** | **4.00** |
| Syn (Acc.) | Passage Ret (en) | 9289 | RWKV6-Finch-7B | **8.00** | **7.00** | 4.00 | 1.50 |
| | | | RWKV6-Finch-7B + OPRM | 5.50 | **7.00** | **7.00** | **3.00** |

Table 16: **LongBench - full results for RWKV-Finch-7B.** We show the results with and without OPRM inference for LongBench (LB) and LongBench-E (0-4k, 4-8k, 8k+). MD-QA, SD-QA, Summ, Syn, Passage Ret (en) stand for MultiDocument-QA, Single Document-QA, Summarization, Synthetic, Passage Retrieval (english). Avg Len is the Average Length in words. RWKV6-Finch-7B + OPRM-Summ uses the summarization technique detailed in Appendix A.3.

