# OpenReview forum: "Overflow Prevention Enhances Long-Context Recurrent LLMs"
_colmweb.org/COLM/2025/Conference — COLM 2025_

### Official Review · Reviewer_HYQK · 2025-04-19

**Rating:** 6
**Confidence:** 5
**Ethics Flag:** 1

**Summary:**

This paper demonstrates that the fixed-size recurrent memory constrains the performance of recurrent LLMs. To alleviate such a problem with additional training, it proposes OPRM that splits long inputs into chunks addressed independently, selects the most relevant chunk by entropy-based or probability-based criteria, and finally gives the decoding token. Besides, OPRM explores an IDK filtering mechanism to prevent LLMs from overconfidence. In order to verify the effectiveness of OPRM, this paper evaluates it on several benchmarks, including LongBench, LongBench V2, and AR tasks.

**Questions To Authors:**

* Line 139: What is the definition of “overflow” here? Does all performance drop indicate memory overflow?

* Line 140: Why does the phenomenon at N=1200 indicate the severity of the said issue in long-context scenarios?

* Line 205: Is the IDK prompt appended?

* Do we re-select chunks at each time step during autoregressive decoding? If so, how does this affect the decoding efficiency?

* What if the context length is not a multiple of the chunk size?

**Reasons To Accept:**

The topic of how to prompt the performance of a recurrent LLM with fixed-size recurrent memory is very interesting. This paper is well organized and written clearly. Its figures and tables are also clear and vivid.

**Reasons To Reject:**

* The Method section is somewhat difficult to understand. I suggest providing the rigorous formulation of the model instead of natural language descriptions.

* Chunk-based inference is not new. OPRM takes a similar strategy as LLM$\times$MapReduce[1].  Then, here are two problems: 1) This paper misses citing LLM$\times$MapReduce and declaring the differences between them. 2) This paper does not compare with LLM$\times$MapReduce.

* It is not clear how this method can be applied to tasks where there is no easy way to partition the prompt into prefix, context, and suffix.

* In experiments, multi-hop benchmarks should be evaluated to verify the effectiveness of OPRM, such as multi-needle NAIH tasks[3] whose needles are scattered in different chunks, code run task in $\infty$Bench[4],  RULER [7].


* Section 3 Problem Investigation seems to not closely to following sections. More details are needed to explain how section 3 guide the design of OPRM. Besides, previous papers also investigate the topic discussed in Section 3 Problem Investigation, please kindly cite them.

* I think the paper should have included sliding window versions of attention [6] and RNNs [2], respectively, as a baselines.

* While it has been mentioned that the proposed method is efficient, there is no experiment to demonstrate this.

* “Acknowledgements” and “Reproducibility” sections should be unnumbered.

* Missing relevant citation [1, 2, 4].



[1] Zhou, Zihan, Chong Li, Xinyi Chen, Shuo Wang, Yu Chao, Zhili Li, Haoyu Wang, Rongqiao An, Qi Shi, Zhixing Tan, Xu Han, Xiaodong Shi, Zhiyuan Liu and Maosong Sun. “LLM$\times$MapReduce: Simplified Long-Sequence Processing using Large Language Models.” (2024). )

[2] Chen, Yingfa, Xinrong Zhang, Shengding Hu, Xu Han, Zhiyuan Liu and Maosong Sun. “Stuffed Mamba: State Collapse and State Capacity of RNN-Based Long-Context Modeling.” ArXiv abs/2410.07145 (2024): n. pag. 2)

[3] https://github.com/gkamradt/LLMTest_NeedleInAHaystack

[4] $\infty$Bench: Extending Long Context Evaluation Beyond 100K Tokens (Zhang et al., ACL 2024)

[5] Li et al. 2024. “FocusLLM: Scaling LLM's Context by Parallel Decoding”.

[6] Xiao et al. 2024. “Efficient Streaming Language Models with Attention Sinks”.

[7] Hsieh et al. 2024. “RULER: What's the Real Context Size of Your Long-Context Language Models?”.

---

> ### Author Response · Authors · 2025-06-03
> **R4 Rebuttal (1/2)**
>
> We thank the reviewer for their detailed and valuable feedback. We address the reviewer’s concerns and questions below:
>
> .
> > OPRM and LLMxMapReduce - Differences and Comparison
>
> **Differences** \
> LLM×MapReduce is an agentic pipeline that handles long contexts by first splitting the context into several chunks for LLMs to process and reason over, and then aggregating their intermediate answers using additional LLM calls to produce the final output.
> In contrast, OPRM is an inference technique for recurrent LLMs that speculatively processes all context chunks and then selects the best one for decoding, all to avoid memory overflows. OPRM requires only a single forward pass.
>
> **Comparison** \
> We applied LLMxMapReduce to Falcon3-Mamba-Inst-7B. While it produced reasonable answers for some queries, it often failed to follow the full set of instructions at different stages of the pipeline. Notably, the LLMxMapReduce paper evaluates Transformer models with a minimum size of 70 billion parameters, whereas Falcon3-Mamba-Inst-7B, at 7 billion parameters, is currently the largest available recurrent LLM. We believe this performance gap is largely due to the significant difference in model scale. It would be interesting to evaluate LLMxMapReduce on larger-scale recurrent LLMs once they become available.
>
> **Efficiency** \
> We report that OPRM is over an order of magnitude faster than LLMxMapReduce: while the latter requires 225.2 seconds per query, OPRM completes a query in 8.6 seconds.
>
>  \
> We will ensure that LLMxMapReduce is properly cited and discuss the differences between the methods, as well as the conditions under which LLMxMapReduce can be applied to recurrent LLMs.
>
>  \
> .
> > multi-hop benchmarks should be evaluated to verify the effectiveness of OPRM
>
> We refer to Section 5.1 (LongBench), where multiple popular multi-hop reasoning benchmarks are evaluated, including: HotpotQA (2-hop), MuSiQue (up to 4-hop), and 2WikiMultihopQA (up to 5-hop). These benchmarks are designed to prevent shortcut-based solutions, and their difficulty has been further increased through the inclusion of distractor passages. \
> Below, we summarize the performance improvements achieved by OPRM on these benchmarks. We report the average results across all four SoTA recurrent LLMs that were tested:
>
> | **HotpotQA** (2 hops) | 0-4k | 4-8k | 8k+ | LB |
> |--|--|--|--|--|
> | Baseline | 27.97 | 21.57 | 17.21 | 22.17 |
> | + OPRM  (Ours) | **38.68** | **34.37** | **36.07** | **35.09** |
> | *Improvement* | 38.3% | 59.3% | 109.7% | 58.3% |
>
>
> | **MuSiQue** (up to 4 hops) | LB |
> |--|--|
> | Baseline | 8.37 |
> | + OPRM  (Ours) | **18.4** |
> | *Improvement* | 119.8% |
>
>
> | **2WikiMultihopQA** (up to 5 hops) | 0-4k | 4-8k | 8k+ | LB |
> |--|--|--|--|--|
> | Baseline | 25.26 | 25.33 | 16.61 | 21.39 |
> | + OPRM  (Ours) | **30.37**  | **28.88** | **27.01** | **25.08** |
> | *Improvement*  | 20.2% | 14.0% | 62.7% | 17.2% |
>
> *LB is LongBench and 0-4k, 4-8k, 8k+ are LongBench-E’s length groups. We note that MuSiQue only has a LB dataset. The baseline models are: Falcon3-Mamba-Inst-7B, Falcon-Mamba-Inst-7B, RecurrentGemma-IT-9B, RWKV6-Finch-7B. ‘Improvement’ shows the additional improvement gained by OPRM. The data is from Tables 9-12 in the Appendix.*
>
> The results above demonstrate that OPRM substantially enhances multi-hop reasoning performance, with improvements exceeding 100% in certain benchmarks across all evaluated SoTA models. These findings show that memory overflows degrade multi-hop reasoning significantly and that OPRM offers an efficient and effective training-free solution.
>
>  \
> To address the concern as thoroughly as possible, we additionally evaluate our method on a large, diverse portion of the proposed InfiniteBench benchmark:
>
> | InfiniteBench | Ret.PassKey | Ret.Number | Ret.KV | En.QA | En.Dia | Code.Debug | Code.Run | Math.Find | Math.Calc |
> |--|--|--|--|--|--|--|--|--|--|
> | Falcon3-Mamba-Inst-7B | 0.0 | 0.0 | 0.0 | 11.0 | 4.0 | 27.1 | 0.0 | 26.9 | 0.0 |
> | + OPRM (Ours) | 100.0 | 100.0 | 31.3 | 23.2 | 8.0 | 24.6 | 0.0 | 33.1 | 0.0 |
>
> Consistent with our earlier findings, OPRM significantly outperforms the baseline. We note that Code.Run is not solved by either model. As reported by InfiniteBench, this synthetic task - which requires computing a complex composition of many functions - remains unsolved even by leading proprietary models such as Claude, Mistral, and Kimi-Chat. The same is for Math.Calc (all proprietary models score 0 on it).
>
>  \
> We conclude that OPRM provides significant improvements in real-world long-context multi-hop tasks, as demonstrated both in our original experiments and in the challenging InfiniteBench evaluation. In the revised version of the paper, we will include these additional results and further emphasize the effectiveness of OPRM for multi-hop reasoning.
>
> .
> > Improving the clarity of the Method section
>
> We will add a formal description of OPRM using LaTeX’s algorithm environment, and compress some of the language descriptions.

---

> > ### Author Response · Authors · 2025-06-03
> > **R4 Rebuttal (2/2)**
> >
> > >  It is not clear how this method can be applied to tasks where there is no easy way to partition the prompt into prefix, context, and suffix.
> >
> > OPRM is primarily designed for instruction-tuned LLMs, which can perform complex tasks described in natural language. In our experiments, we evaluated a wide range of tasks - including: single- and multi-document QA, ICL, code, summarization, document retrieval, and synthetic tasks - and found that the prefix-context-suffix structure is both general and robust. To better address this point, we kindly ask the reviewer to provide examples of such tasks.
> >
> > .
> > > Connection between Section 3 (Problem Investigation) and OPRM; Citations in Section 3
> >
> >
> > An immediate corollary from Section 3 is that the model should not process more information than its recurrent memory capacity allows. If it does, its performance will be severely degraded: Figure 1 (Left, blue curve), Figure 2 (Left).
> >
> > This insight directly motivates OPRM (L161–168): “The core idea behind our method is to chunk the context such that the information content of each chunk does not exceed the model’s limit.” By processing a single, well-selected chunk instead of the entire context, we achieve substantial gains on both real-world and synthetic long-context tasks.
> >
> > In addition, we have cited several works that study the AR capabilities of recurrent architectures [2,3,4]. Please let us know if any relevant work is missing.
> >
> > .
> > > The paper should include sliding window attention and RNNs
> >
> > We refer the reviewer to Section 5.1 (LongBench, LongBench v2), Figure 1 (right), and Tables 1 and 2, where we compare a wide range of recurrent and hybrid architectures, both with and without OPRM. Notably, on LongBench, OPRM inference improves the performance of **RecurrentGemma-IT-9B** (a hybrid of Sliding Window Attention and RG-LRU) **by 50%**, and **RWKV6-Finch-7B** (a leading RNN) **by 51%**. As OPRM is a general approach, it can be applied to many recent models. We will ensure that additional relevant works to which OPRM can be applied are properly cited in the revised version.
> >
> > .
> > > No efficiency experiment
> >
> > See Section B.1 of the Appendix and Table 8, where we evaluate the efficiency of OPRM inference. Our results show that OPRM is faster than vanilla inference, achieving a **25% speedup** at 128k context length. Moreover, we show that the increase in memory is minimal - only a 4.4% overhead at the longest context tested.
> >
> > .
> > > Missing relevant citations
> >
> > We will make sure to insert the experiments and the discussions that were recommended by the reviewer, and will properly cite the related articles.
> >
> > .
> >
> > **Questions**:
> >
> > > L139: What is the definition of “overflow” here? Does all performance drop indicate memory overflow?
> >
> > Yes. We define this explicitly in L116-117: “In the AR task, these overflows are indicated by a decrease in accuracy compared to the initial performance.” \
> > We’re happy to provide additional clarification if needed.
> >
> > .
> > > L140: Why does the phenomenon at N=1200 indicate the severity of the said issue in long-context scenarios?
> >
> > The rationale is that if the model struggles to store and retrieve a modest number of key-value pairs (e.g., 200 pairs in a 1,200-token context), it will likely struggle even more with a larger number in longer contexts (e.g., 5k pairs in 30k tokens). \
> > In real-world tasks, the amount of information typically grows with context length. Therefore, this limitation is expected to manifest in long-context settings - as confirmed by our results in Section 5: Table 5 shows that even a random selection of a single chunk outperforms the baseline by a large margin. Furthermore, In Figure 1 (right) we show that LongBench results improve significantly when processing a single, well selected chunk (OPRM).
> >
> > .
> > > L205: Is the IDK prompt appended?
> >
> > Yes. Usually to the query (which is in the suffix \ prefix).
> >
> > .
> > > Do we re-select chunks at each time step during autoregressive decoding? If so, how does this affect the decoding efficiency?
> >
> > No. The chunk is selected once after the pre-fill phase, and it is the only one used during decoding. \
> > We note that we have experimented with the method that you suggest (select a chunk for each decoded token), but did not see an overall improvement. Given the added complexity of selecting a new chunk after each decoding step, we decided to use ORPM as presented in the paper. Following the reviewers question, we will add the above details to the paper.
> >
> > .
> > > What if the context length is not a multiple of the chunk size?
> >
> > We pad the context so that its length becomes a multiple of the chunk size.
> >
> >  \
> >  \
> > We thank the reviewer for their valuable suggestions and comments, which have helped enrich our paper with additional relevant references, experiments and discussions. If all concerns have been addressed, we kindly request the reviewer to raise their score. Should any concerns remain, we would be happy to address them further.

---

> > > ### Author Response · Authors · 2025-06-03
> > > **R4 Rebuttal - References**
> > >
> > > **References**: \
> > > [1] Language Models are Few-Shot Learners; Brown et. al; NeurIPS 2020 \
> > > [2] Simple linear attention language models balance the recall-throughput tradeoff; Arora et. al; ICLR 2024 \
> > > [3] Zoology: Measuring and Improving Recall in Efficient Language Models; Arora et. al; ICLR 2024 \
> > > [4] Transformers are SSMs: Generalized Models and Efficient Algorithms Through Structured State Space Duality; Dao & Gu; ICML 2024

---

> > > > ### Author Response · Authors · 2025-06-08
> > > > **Author’s Response**
> > > >
> > > > We thank the reviewer again for their constructive feedback, and for raising their score. \
> > > > Please let us know if there are any remaining concerns, as we would be happy to address them within the remaining time.

---

### Official Review · Reviewer_g9uQ · 2025-05-12

**Rating:** 6
**Confidence:** 4
**Ethics Flag:** 1

**Summary:**

Recurrent LLMs are a family of transformer models which all use a fixed-size state to process a long sequence.  The prior context, which may be of unlimited length, is essentially compressed into a fixed-size memory.  It should come as no surprised that as context length increases, the capacity of the fixed-size memory to store the relevant information for later retrieval eventually hits a limit, which the authors call "overflow".

The authors start by measuring overflow using a synthetic associative recall task.

The authors then propose a solution, called OPRM, which is to split the prompt into three parts: [$P$, $C$, $S$]  (Prefix, Context, Suffix).  The context is further divided into a number of chunks $C_1 .. Cn$, and the model is run over each chunk in parallel (with appropriate prefix and suffix -- i.e. [$P$, $C_i$, $S$] for $i \in 1..n$) to calculate a recurrent state.  The best-performing chunk is the selecting according to some mechanism, and used to answer the query.  The authors explore several potential mechanisms for selecting the best chunk.

OPRM boosts performance of recurrent models on long-context tasks, because the chunk size is chosen to be small enough so that a single chunk will not overflow the memory.

**Questions To Authors:**

N/A.

**Reasons To Accept:**

The paper is very well written, and the first few pages which discuss related work and the overflow problem are especially clear.  The authors test their ideas using several different models, on several different datasets, and the experiments seem to be well designed.

UPDATE: score raised after discussion.

**Reasons To Reject:**

Unfortunately, the author's solution to the overflow problem leaves something to be desired, and they fail to discuss the most relevant point of comparison.

"Selecting the best chunk" is just retrieval-augmented-generation (RAG) by another name.  The only difference between what the authors propose, and a typical RAG system, is that the authors do not simply store the text of each chunk.  Instead, they use the recurrent LM to pre-calculate the recurrent state for each chunk.  However, a similar thing can also be done when using conventional RAG in a vanilla transformer; it is possible to pre-calculate the KV-cache for each chunk using sliding-window attention.

The primary research problem in any RAG system is training the retriever.  SOTA work on RAG typically involves training high-quality embeddings for chunks of text, and using a combination of text embeddings and other metrics like BM25 to select the best chunk.  In addition, conventional RAG systems scale naturally from one retrieved chunk to many retrieved chunks; while the same is not true of the author's proposal.

Unfortunately, the author's proposed mechanisms for selecting the best chunk lag far behind SOTA in RAG.  They propose three techniques: an entropy-based measure which IMO is ad-hoc at best, a probability-based measure which looks promising, but performs worse than entropy, and an "I don't know" token.  The IDK token is quite interesting, but does not yield a consistent improvement in all experiments.  The authors do not do a proper comparison against conventional RAG, but I seriously doubt that any of these retrieval techniques would be competitive with SOTA retrievers.

In order for this to be a compelling paper, I would need to see two things.  First, the authors need to discuss RAG systems in the related work section -- they make no mention of RAG anywhere in the paper.  Second, they would need to compare OPRM against conventional RAG.

In my opinion, OPRM could offer a potential advantage over RAG if the size of the recurrent state was smaller than the size of the KV-cache, for a given chunk size.  In other words, the recurrent LM could be used to compress large chunks into a smaller "memories".  However, the authors do not do such a comparison.

---

> ### Author Response · Authors · 2025-05-31
> **R3 Rebuttal**
>
> We thank the reviewer for their detailed and valuable feedback.
>
> The reviewer claims that our method is a form of RAG and points out that we have not compared it with RAG baselines, so the study appears unjustified. We respond as follows:
>
> **(i) Research focus:** \
> The main objective of our research is to study the fundamental limitations of recurrent LLMs in long-context tasks and to explore ways to improve them. We find that recurrent LLMs, when provided with an appropriately sized input segment, can outperform full-context inference. This result is both important and surprising, given that recurrent LLMs are specifically designed for long-sequence processing. Our proposed solution, OPRM, highlights both the limitations of these models and the substantial benefits of context truncation - offering a simple and efficient approach.
>
> **(ii) Methodological differences:** \
> Although there are some similarities, our approach is not conventional RAG. It does not use an external retrieval model, does not involve additional training, and does not follow the typical RAG pipeline of retrieval, prefill, and decode. Instead, it operates in a single forward pass, like a standard LLM. If we adopt the reviewer’s broader definition, where any chunk-selection procedure qualifies as RAG, then the closest analogy would be zero-shot RAG.
>
> That said, to fully address the reviewer’s concerns, we perform additional experiments comparing our method to two zero-shot RAG baselines:
> - DRAGON, which employs a dense retriever designed to generalize in zero-shot settings [1].
> - PRP, a retriever-free method that uses additional forward passes to rank chunks [2].
>
> We evaluate these methods across all four Document QA tasks in LongBench_e (HotpotQA, 2WikiMQA, MultiFieldQA, and Qasper), and report the average performance across all benchmarks. As shown in the table below, OPRM outperforms both methods:
> | Method      | 0-4k      | 4-8k      | 8k+       | Avg       |
> |-------------|-----------|-----------|-----------|-----------|
> | Falcon-Mamba-Inst-7B  | 34.21     | 26.94     | 19.11     | 26.75     |
> | + Dragon      | 35.85     | 30.05     | 32.74     | 32.88     |
> | + PRP         | 36.27  | 32.52    | 34.61   | 34.47     |
> | + OPRM (Ours) |  **37.41**  | **34.49** | **36.25** | **36.05** |
>
> For each RAG method, we tested multiple combinations of num_chunks and chunk_size, ensuring a fair comparison by constraining num_chunks x chunk_size = OPRM_chunk_size. It is also worth noting that PRP's sequential ranking process makes it significantly slower than both OPRM and DRAGON: PRP requires several minutes to process each sample, whereas OPRM and DRAGON take only a few seconds.
>
> These results are consistent with prior findings: RAG-based methods generally do not outperform long-context LLMs on long-context tasks [3,4,5]. To the best of our knowledge, there is no definitive explanation, yet some hypotheses suggest that long-context models may be better at uncovering multi-hop relations, as “…long context(s) are beneficial to increase the recall of incorporating all intermediate hops” [5]. \
> Moreover, the recurrent LLMs used in our work share the same computational complexity as RAG-based solutions. This stands in contrast to the Transformer-based long-context LLMs used in [3,4,5], which incur substantial computational overhead. Taken together, these results and other findings in our paper underscore the potential of long-context recurrent LLMs when combined with overflow prevention techniques such as OPRM. We will incorporate these comparisons and discussion in the revised manuscript.
>
> .
> > The authors need to discuss RAG systems in the related work section.
>
> We will add a subsection to the Related Work section discussing RAG-based systems. This subsection will explicitly cover the RAG pipeline (retrieve, prefill, decode), various types of RAG (external retriever vs. retriever-free, fine-tuning vs. zero-shot, and advanced methods), and the use of RAG in long-context tasks [3,4,5].
>
>
> . \
> We would like to thank the reviewer for raising these important points, which provided valuable perspectives and insights that have helped improve our paper.
>
> . \
> **References:** \
> [1] How to Train Your DRAGON: Diverse Augmentation Towards Generalizable Dense Retrieval; Lin et. al; EMNLP Findings 2023 \
> [2] Large Language Models are Effective Text Rankers with Pairwise Ranking Prompting; Qin et. al; NAACL Findings 2024 \
> [3] Retrieval Augmented Generation or Long-Context LLMs? A Comprehensive Study and Hybrid Approach; Li et. al; EMNLP 2024 \
> [4] LongBench: A Bilingual, Multitask Benchmark for Long Context Understanding; Bai et. al; ACL 2024 \
> [5] Retrieval Meets Long Context Large Language Models; Xu et. al; ICLR 2024

---

> > ### Comment · Reviewer_g9uQ · 2025-06-07
> >
> > I am pleased that the authors had time to run additional experiments to compare OPRM against at least one RAG system, and I appreciate the detailed discussion and references.  I am raising my score.
> >
> > Nevertheless, I still think that the author's proposed technique is a variation on RAG, and many of my original critiques still stand.
> >
> > The authors write that their method: "does not use an external retrieval model.. and does not follow the typical RAG pipeline of retrieval, prefill, and decode."  As I understand it, that's not quite true -- the author's method does in fact follow the typical RAG pipeline of retrieval and decode, it is merely missing the prefill step.  (Instead, it uses the pre-computed recurrent state for the selected chunk.)  I initially dismissed that as an implementation detail, because as I mentioned, one could precompute the KV-cache for each chunk in parallel using sliding-window attention, and thus build a RAG pipeline that closely resembles what the authors propose.  Upon further reflection, however, I may have been too quick to dismiss this difference as an implementation detail, because there are few (or no) existing RAG systems that are actually implemented that way, in which case the author's proposed mechanism is more novel than I thought at first.
> >
> > The authors further write: "RAG-based methods generally do not outperform long-context LLMs [and we don't know why]."  One potential explanation is that if you *don't* pre-compute the KV-cache for each chunk, and instead recompute it during prefill, then each chunk is taken out of context.  This observation is similar to the multi-hop hypothesis, except that the hops need not be long-range.
> >
> > Finally, while it is true that the author's proposed technique "does not use an external retrieval model", I believe that is a major flaw and limitation of their work, since I was unimpressed with the proposed training-free retrieval techniques.
> >
> > In short, I still think this work would be stronger with a better retrieval method, and that it is not as different from RAG as the authors seem to think it is.  However, the discussion of memory overflow is indeed well done and valuable, and I appreciate their efforts to engage constructively with my criticism.

---

> > > ### Author Response · Authors · 2025-06-09
> > > **Author’s Response**
> > >
> > > We thank you for your thorough feedback and for raising your score. \
> > > We appreciate your additional inputs, and will make sure to incorporate them into the revised manuscript.

---

### Official Review · Reviewer_3P6K · 2025-05-25

**Rating:** 7
**Confidence:** 5
**Ethics Flag:** 1

**Summary:**

This paper finds that memory overflows in linear attention limit its associative recall capabilities. To address this, the authors propose parallel chunk-wise processing of input contexts, followed by selecting the chunk most relevant to the target suffix for further autoregressive decoding. The proposed method achieves notable improvements on common long-context benchmarks when applied to Mamba and Mamba2 models.

**Questions To Authors:**

My questions have been included in the above weakness section. I am willing to adjust my scores if my concerns are properly addressed.

**Reasons To Accept:**

1. The idea of parallel chunk-wise processing is novel. Previous works that perform chunk-wise processing typically process each chunk sequentially to save memory, whereas this paper processes all chunks in parallel and demonstrates its effectiveness in avoiding out-of-distribution (OOD) issues and improving long-context performance.

2. The method is particularly interesting because (1) it can enhance parallelism and arithmetic intensity in the memory-bound decoding process, leading to speedup; and (2) it provides a way to extend memory size at inference time without additional training, thereby mitigating OOD issues related to discrepancies between training and test sequence lengths.

3. The proposed method is validated across various long-context tasks and achieves consistent improvements.

4. The paper is clearly written and easy to follow.

**Reasons To Reject:**

1. The connection between the investigation in Section 3 and the proposed method in Section 4 is not strong. The limitations of linear attention in associative recall are well-known and have been studied in works such as DeciMamba and LongMamba. The new insights provided in Section 3 are limited.

2. The definition and detailed mechanism of "memory overflows" require more clarity and a deeper investigation. Currently, this concept is not thoroughly explored in the manuscript.

3. The proposed method is not limited to Mamba or Mamba2 models; it should also be applicable to other linear attention variants, such as DeltaNet and RWKV. More experiments across a broader range of model families are expected.

4. The reported improvements come at the cost of increased memory usage. It would be good to benchmark under the same memory budget, e.g., comparing a smaller model with the proposed technique to a larger model, to see whether the proposed method still yields consistent improvements.

5. During the decoding phase, is chunk-wise processing performed on all previous tokens for each token generation? Or are previous chunks stored as memory for reuse? If the latter, the memory usage may grow linearly with the generation length.

6. It is unclear whether the proposed method can handle scenarios while the required information is spread in different chunks. In such cases, combining some chunks could be better than selecting one chunk.

---

> ### Author Response · Authors · 2025-06-03
> **R2 Rebuttal (1/2)**
>
> We thank the reviewer for the valuable feedback, and address the raised concerns below:
>
> .
> > The limitations of linear attention in AR are well-known and have been studied in works such as DeciMamba and LongMamba. The new insights in Section 3 are limited.
>
> DeciMamba and LongMamba do not study the Associative Recall (AR) task [1,2]. Instead, they analyze long contexts from the view of **length generalization**, focusing on explaining performance degradations that occur when the model processes sequences that are longer than those it was trained on. For clarity, all experiments in Section 3 use contexts that are **shorter** than the training length, hence should not suffer from such issues.
>
> **Section 3 Shows That a More Fundamental Problem Exists:** \
> SoTA recurrent LLMs such as Falcon-Mamba-Inst-7B are: \
> (a) Sufficiently large for good in-context understanding [9] \
> (b) Match Transformer performance on short sequence tasks [13] \
> (c) Do not need to length-generalize in some long-context benchmarks (trained on contexts *longer* than the ones in the benchmark) \
> (d) Feature a significantly large hidden state - which should correlate with increased recurrent memory capacity [3-6].
>
> Yet, despite all of the above, these models still *underperform* on real-world long-context tasks. \
> To investigate this, we turn to the AR task and show that a more fundamental problem exists (L114–115). \
> Surprisingly, under these “good conditions”, the AR score is much lower than expected: the model cannot reliably store more than 40 key-value pairs (Figure 1, left), and its performance degrades sharply under memory overload. This suggests that while recurrent LLMs trained via next-token prediction may develop some form of recurrent memory, **its capacity falls far short of both the demands of long-context tasks and the model’s actual memorization potential.**
>
> Indeed, AR has been used in prior work. However, to the best of our knowledge, we are the first to apply it to large-scale, long-context recurrent models and use it to identify memory overflows as a fundamental limitation in long-context settings.
>
> .
> > The proposed method is not limited to Mamba; a broader range of model families are expected.
>
> We refer to Section 5.1 (LongBench, LongBench v2), Figure 1 (Right), and Tables 1&2, where we evaluate a variety of recurrent and hybrid architectures. Notably, on LongBench, OPRM inference improves **RWKV6-Finch-7B by 51%**. It also improves **RecurrentGemma-IT-9B** (a hybrid of Sliding Window Attention and RG-LRU) **by 50%**. These results (via RWKV, Griffin, Falcon-Mamba and Falcon3-Mamba) show that OPRM generalizes well beyond Mamba, offering substantial gains across diverse recurrent and hybrid architectures.
>
> .
> > It is unclear whether OPRM can handle scenarios where the required information is spread in different chunks. Combining chunks could be better.
>
> To address this concern, we conduct the following experiment: \
> We use the OPRM pipeline to first identify top-k (smaller) relevant chunks and then evaluate them together with an additional forward pass. We refer to this method as CC (Combined Chunks). We evaluated CC’s performance across all 4 Document QA tasks in LongBench_e and report the average scores:
>
> | Method | 0-4k | 4-8k | 8k+ | Avg |
> |--|--|--|--|--|
> | Baseline | 34.21 | 26.94 | 19.11 | 26.75 |
> | + CC | **40.27** | **35.56** | 35.98 | **37.27** |
> | + OPRM | 37.41 | 34.49 | **36.25** | 36.05 |
>
> *Falcon-Mamba-Inst-7B is the baseline (vanilla inference). ‘+’ indicates the inference algorithm. The 4 datasets are HotpotQA, 2WikiMQA, MultiFieldQA, and Qasper. The total length of the CC chunks is constrained to equal one OPRM chunk. We experimented with multiple combinations of chunk length and number, and used the best parameters for testing.*
>
> \
> We see that while CC is beneficial for short contexts (0-4k), it offers no clear advantage over OPRM in longer contexts (4-8k and 8k+). Moreover, it adds computational and algorithmic overheads. \
> One possible explanation is that longer contiguous contexts help the model uncover multi-hop relations [10], as some reasoning steps depend on access to previously seen information. In such cases, given the same length budget, selecting longer segments may be more effective than using multiple smaller chunks: when a chunk is small, its relevance is more likely to be biased toward surface-level similarity with the query. In contrast, longer segments include additional context that can improve the relevance estimation.
>
> We conclude that cross-chunk information fusion warrants further investigation (L333-334). For example, a system-oriented approach, such as an overflow-aware variant of Bahdanau Attention [8] or hierarchical state processing, may hold significant potential. We will add the above experiment and discussion to the revision of the paper.

---

> > ### Author Response · Authors · 2025-06-03
> > **R2 Rebuttal (2/2)**
> >
> > > The connection between Section 3 (problem investigation) and Section 4 (method) is not strong.
> >
> > An immediate corollary from Section 3 is that the model should not process more information than its recurrent memory capacity allows. If it does, its performance will be severely degraded, as demonstrated in Figure 1 (Left, blue curve) and Figure 2 (Left).
> >
> > This insight directly motivates our method (L161–168): “The core idea is to chunk the context such that the information content of each chunk does not exceed the model’s limit.” By processing a single, well-selected chunk instead of the entire context, we achieve substantial gains on both real-world and synthetic tasks. This simple modification highlights both the limitations of current models and the strong benefits of context truncation.
> >
> > .
> > > The definition and detailed mechanism of "memory overflows" requires more clarity and a deeper investigation.
> >
> > We distinguish between two related but distinct contributions: (1) identifying the phenomenon of memory overflows as a major bottleneck in long-context recurrent LLMs, and (2) investigating the underlying circuit-level mechanisms. \
> > While both are important, we primarily focus on the former for these reasons:
> >
> > **(i) A step toward mechanistic understanding.** \
> > Seminal works in the field, such as Mamba2 [5], remain uncertain about the exact factors that contribute to improved recurrent memory capacity: \
> > *“… we are not sure which aspect of the architecture is the predominant factor (for improving AR performance in 2-layer toy models), which remains a question to explore in future work.”* \
> > This problem is even more challenging for large-scale recurrent LLMs, as they are not explicitly trained for AR and have much more parameters. \
> > With that being said, identifying that memory overflows limit long context recurrent LLMs is a necessary first step toward uncovering its underlying mechanism. In other domains, similar observations have led to the discovery of concrete circuits. For example, in-context learning motivated the identification of induction heads [11], and strong arithmetic abilities led to the discovery of The “Clock” and “Pizza" mechanisms [12].
> >
> > **(ii) Isolating the long-context bottleneck.** \
> > While previous work has examined the limitations of recurrent LLMs through the lens of effective receptive field or poor length generalization [1,2], our empirical analysis points to a different root cause. By evaluating SoTA recurrent LLMs in settings that explicitly rule out explanations (a)-(d) (as discussed in the first response), we demonstrate that memory overflows constitute a fundamental limitation for efficient long-context processing.
> >
> >  \
> > For clarity, we will revise line 116 to define memory overflow as follows: \
> > *Memory overflows occur when a model fails to store or retrieve relevant information from its context due to state overloading. Specifically, when less information (e.g., fewer facts) is provided, the recurrent model can retain and retrieve it successfully. However, as more information is introduced, the model’s internal state becomes saturated, leading to retrieval failures. Memory overflow is a fundamental limitation of recurrent models that extends beyond other ones such as sequence length or model size.*
> >
> > We note that this general definition encompasses the task-specific definition for AR provided in line 117: \
> > *“In the AR task, overflows are indicated by a decrease in accuracy compared to the initial performance.”*
> >
> > .
> > > benchmarking memory usage under the same memory budget
> >
> > We respectfully disagree on the necessity of this test. In the most extreme case we evaluated, OPRM requires only 4.4% more memory than the baseline (Table 8). Given that the tested model has 7 billion parameters, this would correspond to comparing it with a 6.69 billion parameter model. However, LLMs are not typically trained at such fine-grained parameter scales, and a 6.69B model is widely considered equivalent to a 7B model in practice.
> >
> > .
> > > During the decoding phase, is chunk-wise processing performed on all previous tokens for each token generation? Or are previous chunks stored as memory for reuse? If the latter, the memory usage may grow linearly with the generation length.
> >
> > No, chunk-wise processing is performed only during the pre-fill phase (Figure 3). After this phase, a single chunk is selected, and we decode all future tokens with it. Previous chunks are not stored for reuse.
> >
> >  \
> >  \
> > We would like to thank the reviewer for their valuable comments and suggestions, which have helped improve our paper. If all concerns have been addressed, we kindly request the reviewer to raise their score. Should any concerns remain, we would be happy to address them further.

---

> > > ### Author Response · Authors · 2025-06-03
> > > **R2 Rebuttal - References**
> > >
> > > **References:**
> > >
> > > [1] DeciMamba: Exploring the Length Extrapolation Potential of Mamba; Ben-Kish et. al; ICLR 2025 \
> > > [2] LongMamba: Enhancing Mamba's Long Context Capabilities via Training-Free Receptive Field Enlargement; Ye et. al; ICLR 2025 \
> > > [3] Simple linear attention language models balance the recall-throughput tradeoff; Arora et. al; ICLR 2024 \
> > > [4] Zoology: Measuring and Improving Recall in Efficient Language Models; Arora et. al; ICLR 2024 \
> > > [5] Transformers are SSMs: Generalized Models and Efficient Algorithms Through Structured State Space Duality; Dao & Gu; ICML 2024 \
> > > [6] Linear Transformers Are Secretly Fast Weight Programmers; Schlag et. al; PMLR 2021 \
> > > [8] Neural Machine Translation by Jointly Learning to Align and Translate; Bahadanau et. al; ICLR 2015 \
> > > [9] Language Models are Few-Shot Learners; Brown et. al; NeurIPS 2020 \
> > > [10] Retrieval Meets Long Context Large Language Models; Xu et. al; ICLR 2024 \
> > > [11] In-context Learning and Induction Heads; Olsson et. al; Anthropic 2022 \
> > > [12] The Clock and the Pizza: Two Stories in Mechanistic Explanation of Neural Networks; Zhong et. al; NeurIPS 2023 \
> > > [13] Falcon Mamba: The First Competitive Attention-free 7B Language Model; Zuo et. al; Falcon Team, 2024 \

---

> > > > ### Author Response · Authors · 2025-06-08
> > > > **Kind Reminder: Discussion Deadline Approaching**
> > > >
> > > > We thank the reviewer again for their feedback. \
> > > > As the discussion deadline approaches, we would greatly appreciate knowing if there are any remaining concerns, as we would be more than happy to address them in the remaining time.

---

> > > > > ### Comment · Reviewer_3P6K · 2025-06-09
> > > > >
> > > > > Thank the authors for their response, especially for the clarification on the associative recall presented in this work, which makes the analysis different from existing works. Most of my concerns have been addressed, and I will raise my score to 7.

---

> > > > > > ### Author Response · Authors · 2025-06-09
> > > > > > **Author’s Response**
> > > > > >
> > > > > > We would like to thank you again for the constructive feedback and appreciate your decision to raise your score.

---

### Official Review · Reviewer_jWUi · 2025-05-26

**Rating:** 6
**Confidence:** 4
**Ethics Flag:** 1

**Summary:**

This paper investigated the performance of leading recurrent LLMs with fixed-size recurrent memory on long-context sequence processing tasks. Concretely, the experimental results revealed that these recurrent LLMs suffer from chronic underuse of long-context information.

Based on the empirical observations, the authors further proposed a chunk-based inference method to prevent information overflow in recurrent LLMs. This method first splits the context into multiple segments with fixed length. Then, it applied the prefix-context-query to each segment and collect all the generated answers. At last, the method uses a selective decoding method to select the most relevant segment for the final output. Experiments on LongBench v1 and v2 demonstrate the improvements of the proposed chunk-based inference method outperforms the standard inference one. This raises a crucial question about context utilization of recurrent LLMs.

**Reasons To Accept:**

1. The question in this paper is well-motivated and valuable

2. The experiments are well-conducted

3. The proposed inference method outperforms the standard decoding method

**Reasons To Reject:**

1. All experiments of AR were conducted under one sequence length with 1,200 tokens. It is recommended to evaluate AR with different sequence lengths.

2. It is recommended to control the position of the queried KV pair in the AR task to better understand the recurrent memory failure problem.

---

> ### Author Response · Authors · 2025-05-31
> **R1 Rebuttal**
>
> We thank the reviewer for their valuable comments and suggestions.
>
> .
> > All experiments of AR were conducted under one sequence length with 1,200 tokens. It is recommended to evaluate AR with different sequence lengths.
>
> Following the reviewer’s suggestion, we extend the AR experiment to multiple sequence lengths: 300, 600, 1200, 2400, 4800. The results can be found here: https://bashify.io/i/xsnIwA \
> Our findings indicate that memory capacity is not particularly sensitive to the overall context length. \
> Similar to the original 1,200-token setting, performance starts at around 75% accuracy for 10 key-value pairs (facts), and gradually declines as the number of facts increases, eventually converging toward 0% accuracy. \
> Note that each fact is 6 tokens long, therefore for sequence length L=300 we can only test a maximum of 50 facts and for L=600 a maximum of 100 facts.
>
> .
> > It is recommended to control the position of the queried KV pair in the AR task to better understand the recurrent memory failure problem.
>
> In response to the reviewer’s recommendation, we analyze the position of successfully retrieved key-value (KV) pairs. The results can be found here:  https://bashify.io/i/R2qtbs \
> Each plot presents a normalized histogram of the locations of successfully retrieved KV pairs (facts). In all measurements we use 10 equally spaced bins to aggregate the samples, and average over 5 different seeds. When the number of facts is small, the distribution appears relatively uniform, which corresponds with the high AR accuracy. Interestingly, as the number of facts increases, we observe a U-shaped recall pattern, resembling the lost-in-the-middle phenomenon reported in [1]. \
> We conclude that while recall performance degrades across all positions, the middle portions of the context suffer the most. Since this phenomenon is also observed in Transformer-based LLMs, we suspect that positional relevance may be influenced by properties of the data itself rather than the model's architecture. This noteworthy finding warrants further investigation in future work.
>
> . \
> We will add both experiments to the revised manuscript. We thank the reviewer for suggesting these experiments, which offer deeper insight into the behavior of the recurrent memory in recall-intensive tasks, and help improve the paper.
>
> \
> **References**: \
> [1] Lost in the Middle: How Language Models Use Long Contexts; Liu et. al; TACL 2023

---

> > ### Author Response · Authors · 2025-06-08
> > **Kind Reminder: Discussion Deadline Approaching**
> >
> > We thank the reviewer again for their feedback. \
> > As the discussion deadline approaches, we would greatly appreciate knowing if there are any remaining concerns, as we would be more than happy to address them in the remaining time.

---

### Author Response · Authors · 2025-06-05
**Summary of Revisions and More InfiniteBench Results**

We sincerely thank all reviewers for their detailed and constructive feedback. \
To thoroughly address the concerns raised, we conducted new experiments and made several updates to the paper. A summary of the key additions is provided below:

1. **Comparison to RAG baselines**: \
OPRM outperforms both RAG baselines tested. It achieves this while providing highly efficient inference and a simpler algorithmic framework.

2. **Highlighting OPRM’s gains on popular multi-hop reasoning benchmarks**: \
OPRM significantly improves performance across all benchmarks for a variety of SoTA recurrent and hybrid LLMs. In some cases, it even **doubles** the baseline’s score.


3. **Additional benchmark - InfiniteBench**: \
We report significant gains on the challenging InfiniteBench benchmark, which involves ultra-long contexts (100k–200k tokens). Below are the full InfiniteBench results (added En.Sum and En.MC). The new results are consistent with the previously reported trend - OPRM inference significantly outperforms the baseline:

| InfiniteBench | Ret.PassKey | Ret.Number | Ret.KV | En.QA | En.Dia | Code.Dbg | Code.Run | Math.Calc | Math.Find | En.Sum | En.MC | **Avg** |
|--|--|--|--|--|--|--|--|--|--|--|--|--|
| Falcon3-Mamba-Inst-7B | 0.0          | 0.0          | 0.0        | 11.0      | 4.0      | **27.1** | 0.0 | 26.9      | 0.0 | 20.1      | 45.4      | 12.2      |
| + OPRM (Ours)                | **100.0** | **100.0** | **31.3** | **23.2** | **8.0** | 24.6      | 0.0 | **33.1** | 0.0 | **22.1** | **59.4** | **36.5** |


4. **Initial exploration of cross-chunk information fusion**: \
We show that for long sequences, selecting the top-k relevant chunks offers no advantage over using a single long chunk under the same length budget. This initial result motivates future work on integrating overflow-aware cross-chunk fusion into the architecture.

5. **Additional “Problem Investigation” experiments (Section 3 and Appendix)**:
- Memory capacity is not sensitive to sequence length.
- Facts at the beginning and end of the context are less prone to overflows, indicating positional sensitivity.

6. **Additional discussions and clarifications**:
- RAG systems are now covered in the Related Work section.
- Agentic frameworks for long-context processing such as LLMxMapReduce.
- The Method section has been clarified.
- The definition of memory overflow has been refined for clarity.

\
Following the reviews, the revised manuscript presents a more comprehensive and clearer account of our contributions while addressing reviewer concerns. \
Given the above, we would appreciate it if you would reconsider our score. \
In addition, if there are still any remaining concerns, we would be more than happy to address them. \
We once again thank the reviewers for their valuable insights and suggestions.

---

### Decision · Program_Chairs · 2025-07-08

**Decision:**

Accept

**Comment:**

This paper investigates the ability of recurrent language models to process context. It identifies key limitations of recurrent models and proposes a novel method (OPRM) for inference that strongly improves the performance of several recurrent LLMs on challenging long-context benchmarks. After an active discussion period, all reviewers recommend acceptance.